# The GNU subunit of PNG kinase, the developmental regulator of mRNA translation, binds BIC-C to localize to RNP granules

Emir E Avilés-Pagán[1,2†], Masatoshi Hara[2‡], Terry L Orr-Weaver[1,2*]

[1]Department of Biology, MIT, Cambridge, United States; [2]Whitehead Institute for Biomedical Research, Cambridge, United States

**Abstract** Control of mRNA translation is a key mechanism by which the differentiated oocyte transitions to a totipotent embryo. In *Drosophila*, the PNG kinase complex regulates maternal mRNA translation at the oocyte-to-embryo transition. We previously showed that the GNU activating subunit is crucial in regulating PNG and timing its activity to the window between egg activation and early embryogenesis (Hara et al., 2017). In this study, we find associations between GNU and proteins of RNP granules and demonstrate that GNU localizes to cytoplasmic RNP granules in the mature oocyte, identifying GNU as a new component of a subset of RNP granules. Furthermore, we define roles for the domains of GNU. Interactions between GNU and the granule component BIC-C reveal potential conserved functions for translational regulation in metazoan development. We propose that by binding to BIC-C, upon egg activation GNU brings PNG to its initial targets, translational repressors in RNP granules.

*For correspondence:
weaver@wi.mit.edu

Present address: †Morrison and Foerster, San Francisco, United States; ‡Graduate School of Frontier Biosciences, Osaka University, Suita, Japan

Competing interests: The authors declare that no competing interests exist.

## Introduction

The transition from oocyte to embryo marks the onset of development for most metazoans. Egg activation triggers this transition and results in, amongst other changes, completion of the meiotic program in the oocyte and restoration of a totipotent cell state (*Avilés-Pagán and Orr-Weaver, 2018*; *Krauchunas and Wolfner, 2013*). This transition is regulated exclusively by post-transcriptional mechanisms, as it occurs in the absence of new transcription and depends on translational control of stockpiled maternal mRNAs as well as proteolysis and posttranslational modification of proteins. These maternal mRNAs also support early embryogenesis until the onset of zygotic transcription. The stockpiles of maternal mRNAs and absence of transcriptional input at the oocyte-to-embryo transition provide the opportunity to elucidate conserved mechanisms regulating mRNA translation during a cell state transition.

In *Drosophila*, the PNG kinase complex is a master regulator of mRNA translation at the oocyte-to-embryo transition. The active kinase requires the PNG catalytic subunit, a kinase with a preference for threonine (*Hara et al., 2018*), and the PLU and GNU activating subunits (*Lee et al., 2003*). Initially shown to control translation of a few targets (*cyclins A* and *B*, *smg*) (*Tadros et al., 2007*; *Vardy and Orr-Weaver, 2007*), recent analyses of global translation revealed that around 90% of all transcripts present in the activated egg are completely or partially dependent on PNG to either activate or inhibit their translation following egg activation (*Kronja et al., 2014a*).

Following egg activation, PNG phosphorylates the translational regulators TRAL, ME31B, and BIC-C amongst other regulators of mRNA translation (*Hara et al., 2018*). Phosphorylation by PNG is thought to inhibit the translational repressor activity of TRAL and PUM (*Hara et al., 2018*; *Vardy and Orr-Weaver, 2007*), suggesting a key mechanism by which PNG directly alters

translation through changing the activity of translational repressors. PNG activity also is required for the degradation of ME31B and TRAL at the onset of the maternal-to-zygotic transition (MZT) later in embryogenesis (*Wang et al., 2017*). In addition, translation of PNG mRNA targets such as *smg* is required for translational repression and clearance of maternal transcripts at the MZT (*Tadros et al., 2007*), one example of indirect regulation of the maternal mRNA pool by PNG. PNG can affect both polyadenylation and deadenylation of maternal mRNAs (*Eichhorn et al., 2016*). Thus, the PNG complex regulates maternal mRNA translation through various downstream mechanisms.

PNG kinase complex activity is limited to a narrow developmental window. Crucial to this regulation is the coordinated binding of the subunits of the complex (*Hara et al., 2017*). Whereas PNG and PLU are bound in mature oocytes, phosphorylation of GNU by CDK1/CYCB inhibits its binding to the complex (*Hara et al., 2017*). Thus, the complex is inactive in mature oocytes. Upon egg activation, GNU is dephosphorylated, and it is then able to bind and activate the kinase activity of the complex. The complex is once again inactivated following egg activation by PNG-dependent degradation of GNU, thereby limiting its activity to the oocyte-to-embryo transition. It remains unknown if, in addition to its role in the activation of PNG, GNU has additional roles in mature oocytes or during egg activation.

Cytoplasmic RNP granules are observed in the oocytes of many metazoans (*Schisa, 2012*). RNP granules are membrane-less organelles that regulate different aspects of mRNA metabolism in cells (*Decker and Parker, 2012*). In addition to their roles in oocytes, RNP granules or P-bodies are also present in other cell types, such as neurons, where they have been implicated in the control of rapid cell state transitions (*De Graeve and Besse, 2018*). In oocytes, RNP granules can be observed as large granular structures by immunostaining of resident proteins or in situ hybridization to localized mRNAs (*Kato and Nakamura, 2012*; *Noble et al., 2008*; *Schisa, 2012*). Oocyte RNP granules, similar to their counterparts in other cells, are thought to play roles in regulating the localization and translation of mRNA targets (*Kato and Nakamura, 2012*; *Schisa, 2012*). In *Drosophila*, egg activation triggers the disassembly of these 'oocyte' granules (*Weil et al., 2012*), and they reform as 'embryonic' RNP complexes in early embryos (*Wang et al., 2017*), consistent with roles for these complexes in the regulation of maternal mRNAs at this time.

Along with the RNA components of RNP granules, conserved protein components of P-bodies have been identified (*Hubstenberger et al., 2017*; *Standart and Weil, 2018*), with DDX6 (*Drosophila* ME31B) and LSM14 (aka RAP55 and *Drosophila* TRAL) being key. In *Drosophila* oocytes, both TRAL and ME31B are localized in RNP granules (*Nakamura et al., 2001*; *Wilhelm et al., 2005*). Another translational repressor, BIC-C, also localizes to RNP granules in oocytes of some species, such as in *Drosophila* (*Kugler et al., 2009*). It is hypothesized that the recruitment of these proteins to RNP granules is reflective of the role of these structures in mRNA storage and translational control both in the quiescent oocyte and during egg activation. The change observed as oocyte RNP complexes are disassembled upon egg activation and give way to the formation of embryonic RNP complexes may reflect developmental changes in mRNA translation (*Hubstenberger et al., 2013*; *Kato and Nakamura, 2012*; *Wang et al., 2017*). A complete understanding of which proteins localize and function within oocyte RNP granules, as well as the different roles played by these complexes during the oocyte-to-embryo transition, is lacking.

Here, we examine the regulation of the GNU activating subunit of the PNG kinase complex in mature *Drosophila* oocytes. We found by immunoprecipitation analysis, localization studies, and genetic interactions that GNU is a component of RNP granules present in mature *Drosophila* oocytes. Furthermore, we define roles for the sterile-alpha-motif (SAM) domain of GNU and the CDK1 phosphorylation sites in the interactions of GNU with RNP granules and PNG regulation. Finally, we discuss new models of activation of the PNG complex and control of mRNA translation during egg activation based on these findings.

## Results

### GNU interacts with RNP components in mature oocytes

GNU is not in a complex with PNG in mature oocytes, raising the question of whether GNU alone plays roles prior to egg activation. As a first step to address this, we identified proteins that potentially interact with GNU by immunoprecipitation followed by mass spectrometry. We decided to use a previously reported functional GNU-GFP fusion protein expressed in *gnu-gfp* transgenic lines to pull-down GNU through the GFP tag (***Hara et al., 2017***). In these transgenic lines, GNU-GFP is expressed under endogenous regulatory elements, and GNU-GFP is expressed at levels comparable to wild-type, untagged GNU (***Hara et al., 2017***). In our experiments, however, the transgenes were crossed into a *gnu*$^{305}$ null mutant background, so the only form of GNU present was the GNU-GFP fusion.

Mature oocytes were isolated and extracts were prepared for incubation with anti-GFP beads. Following the pull-down, we analyzed the eluate samples by mass spectrometry to identify potential interactors with GNU, also performing pull-downs from a no GFP transgene control. In addition, immunoprecipitations were done from extracts prepared from mature oocytes expressing H2Av-GFP to control for interactions specific to the GFP tag. Each pull-down was performed in at least two biological replicates, and to be considered valid hits, we required that any given protein was represented in both replicates and enriched at least sevenfold in GNU-GFP precipitates over the no transgene control (***Table 1***, ***Figure 1—figure supplement 1***, and ***Supplementary file 1***). In the experiment using H2Av-GFP to control for interactions with GFP, we required that proteins be enriched at least fourfold in GNU-GFP versus H2Av-GFP pull-downs (***Figure 1—figure supplement 2A***).

Given PNG's key role in controlling maternal mRNA translation at egg activation (***Kronja et al., 2014a***), it was striking that the most enriched proteins recovered were translational repressors. The translational repressor BIC-C had the most enriched interaction with GNU (***Table 1***, ***Figure 1—figure supplement 1***, and ***Supplementary file 1***). The translational repressors TYF and YPS were also significantly enriched in GNU-GFP pull-downs. Interestingly, BIC-C and YPS form a complex with TRAL, ME31B, and CUP during oogenesis (***Chicoine et al., 2007***; ***Kugler et al., 2009***; ***Wilhelm et al., 2005***), so the immunoprecipitation of these proteins by GNU suggests an interaction of GNU with this complex. Although not significantly enriched under our stringent criteria, a noticeable spectrum count was observed for ME31B in GNU-GFP pull-downs (***Table 1***). Furthermore, the

**Table 1.** Interactors with GNU in mature oocytes identified through IP-MS.

| | Total spectrum count | | | | | | | | | | | | |
|---|---|---|---|---|---|---|---|---|---|---|---|---|---|
| | No GFP | | GNU$^{WT}$-GFP | | | GNU$^{9A}$-GFP | | GNU$^{\Delta SAM}$-GFP | | H2Av-GFP | | | GNU$^{WT}$-GFP (+RNase A) |
| **Identified protein** | **Rep 1** | **Rep 2** | **Rep 1** | **Rep 2** | **Rep 3** | **Rep 1** | **Rep 2** | **Rep 1** | **Rep 2** | **Rep 1** | **Rep 2** | **Rep 3** | |
| GFP | – | – | 319 | 376 | 307 | 62 | 52 | 353 | 520 | 358 | 49 | 31 | 467 |
| GNU | – | – | 179 | 178 | 220 | 81 | 90 | 152 | 109 | 3 | – | – | 205 |
| BIC-C | – | – | 92 | 13 | 73 | 5 | – | – | – | – | – | – | 90 |
| TYF | – | – | 64 | 6 | 11 | 9 | – | 45 | 5 | 11 | 3 | – | 38 |
| YPS | – | – | 40 | 11 | 23 | 10 | – | 38 | 2 | 28 | – | – | – |
| TWS | – | – | 39 | 9 | 22 | 10 | 3 | 92 | 14 | – | – | – | 32 |
| TRAL | 3 | 5 | 37 | 7 | 17 | 5 | 5 | 22 | 2 | 20 | 2 | – | 11 |
| PP2A-29B | – | – | 27 | 4 | 30 | 8 | 4 | 52 | 7 | 5 | – | – | 24 |
| CUP | – | – | 20 | 5 | 7 | 7 | 4 | 9 | – | 4 | – | – | 9 |
| ATX-2 | – | – | 20 | 6 | 9 | – | – | 23 | 6 | 6 | – | – | 16 |
| ME31B | 2 | 2 | 11 | 3 | 6 | 6 | 8 | 6 | 2 | 9 | – | – | 6 |
| MTS | – | – | 10 | – | 10 | 3 | – | 42 | 5 | – | – | – | 8 |
| RPL36A | – | – | 7 | 5 | 5 | – | – | 3 | 2 | – | – | – | 7 |

interaction with BIC-C was not affected by the treatment of extracts with RNase A, indicating that this interaction does not depend on RNA (*Table 1*).

ME31B, TRAL, and BIC-C are PNG phosphorylation substrates, and genetic data are consistent with a functional interaction between PNG and TRAL (*Hara et al., 2018*). We confirmed the interaction of GNU-GFP with BIC-C and ME31B by immunoprecipitation/immunoblot analysis (*Figure 1A*, lanes 2 and 7, *Figure 1—figure supplement 3*). Both BIC-C and ME31B were detected in GNU-GFP pull-down samples, in accordance with the mass spectrometry analysis. These interactions are not indirectly mediated by RNA, as they were unaffected by treatment with RNase A (*Figure 1A*, compare lanes 7 and 11). Although pull-downs were performed using an available YPS-GFP line, we were not able to confirm the interaction between YPS and GNU through immunoprecipitation/immunoblot analysis.

Another major group of proteins isolated after immunoprecipitation were components of the PP2A phosphatase complex, MTS, PP2A-29B, and TWS. These proteins were highly enriched in GNU-GFP pull-downs over no GFP control (*Table 1*, *Figure 1—figure supplement 1*, and *Supplementary file 1*). Moreover, the interaction with TWS was significantly higher in GNU-GFP compared to the H2Av-GFP control (*Table 1*). Given that GNU is known to become dephosphorylated at egg activation (*Hara et al., 2017*), the presence of these phosphatase subunits in a complex with GNU raises a potential role for this complex in the dephosphorylation of GNU. Although this is an intriguing possibility, for this study, we prioritized analysis of the relationship between GNU and BIC-C, ME31B, and TRAL.

Because CDK1 phosphorylation of GNU changes its association with PNG (*Hara et al., 2017*), we investigated whether its interaction with BIC-C and ME31B was affected by the phosphorylation state of GNU in mature oocytes (*Krauchunas et al., 2012*). We performed pull-downs from extracts of mature oocytes expressing a *gnu$^{9A}$-gfp* transgene. The GNU$^{9A}$ mutant has all nine CDK1 phosphorylation sites mutated to alanine (*Figure 1B*), inhibiting phosphorylation by CDK1 and mimicking the dephosphorylated state of GNU. GNU$^{9A}$-GFP is still able to bind and activate PNG, and confers partial GNU function (*Hara et al., 2017*). We found that most interactions with GNU were unaffected by the GNU$^{9A}$-GFP mutant (*Table 1*, and *Figure 1—figure supplement 2B*). However, BIC-C was significantly lower in GNU$^{9A}$-GFP than in GNU$^{WT}$-GFP precipitates, suggesting that the interaction of GNU with BIC-C depends on the CDK1 phosphorylation state of GNU. The reduced interaction of GNU$^{9A}$-GFP with BIC-C was confirmed by immunoprecipitation/immunoblot analysis (*Figure 1A*, lanes 3 and 8, *Figure 1—figure supplement 3*). It is possible that the phosphorylation state of GNU affects the strength of the interaction with BIC-C, leading to the observed reduction in BIC-C levels in GNU$^{9A}$-GFP precipitates.

## GNU interaction with BIC-C is dependent on the SAM domain

From the amino acid sequence, the sole recognizable domain in GNU is a SAM domain at its C-terminus. The N-terminal region is predicted to be intrinsically disordered, and it contains most of the CDK1 phosphorylation sites (*Hara et al., 2017*). SAM domains can mediate both protein-protein and protein-RNA interactions (*Green et al., 2003*; *Kim and Bowie, 2003*). We thus examined whether the SAM domain of GNU is needed for the protein interactions we identified. We performed immunoprecipitation/mass spectrometry analyses of extracts from mature oocytes expressing a *gnu$^{\Delta SAM}$-gfp* transgene. This transgene expresses GNU$^{\Delta SAM}$-GFP, which carries a deletion of the SAM domain (*Figure 1B*). We found that most interactions were unaffected by deletion of the SAM domain, with only the interaction of GNU with BIC-C being blocked (*Figure 1—figure supplement 2C*). We confirmed the requirement of the SAM domain for the interaction between GNU and BIC-C by immunoprecipitation/immunoblot analysis. BIC-C immunoprecipitated with GNU$^{WT}$-GFP but not GNU$^{\Delta SAM}$GFP (*Figure 1A*, lanes 4 and 9, *Figure 1—figure supplement 3*), consistent with the SAM domain of GNU being necessary for the interaction with BIC-C, an interaction that we showed above does not require RNA. In the immunoprecipitation/immunoblot experiment, the amount of ME31B immunoprecipitated with GNU$^{\Delta SAM}$GFP appears less than with GNU$^{WT}$-GFP. Because we did not detect a statistically significant enrichment of ME31B in the pull-down/mass spec experiments, we could not confirm that the interaction between ME31B and GNU is affected by deletion of the SAM domain by this independent method.

The observed association between GNU and BIC-C in vivo prompted us to test whether in vitro this could be a direct interaction. GNU and BIC-C were found previously to interact in yeast two-

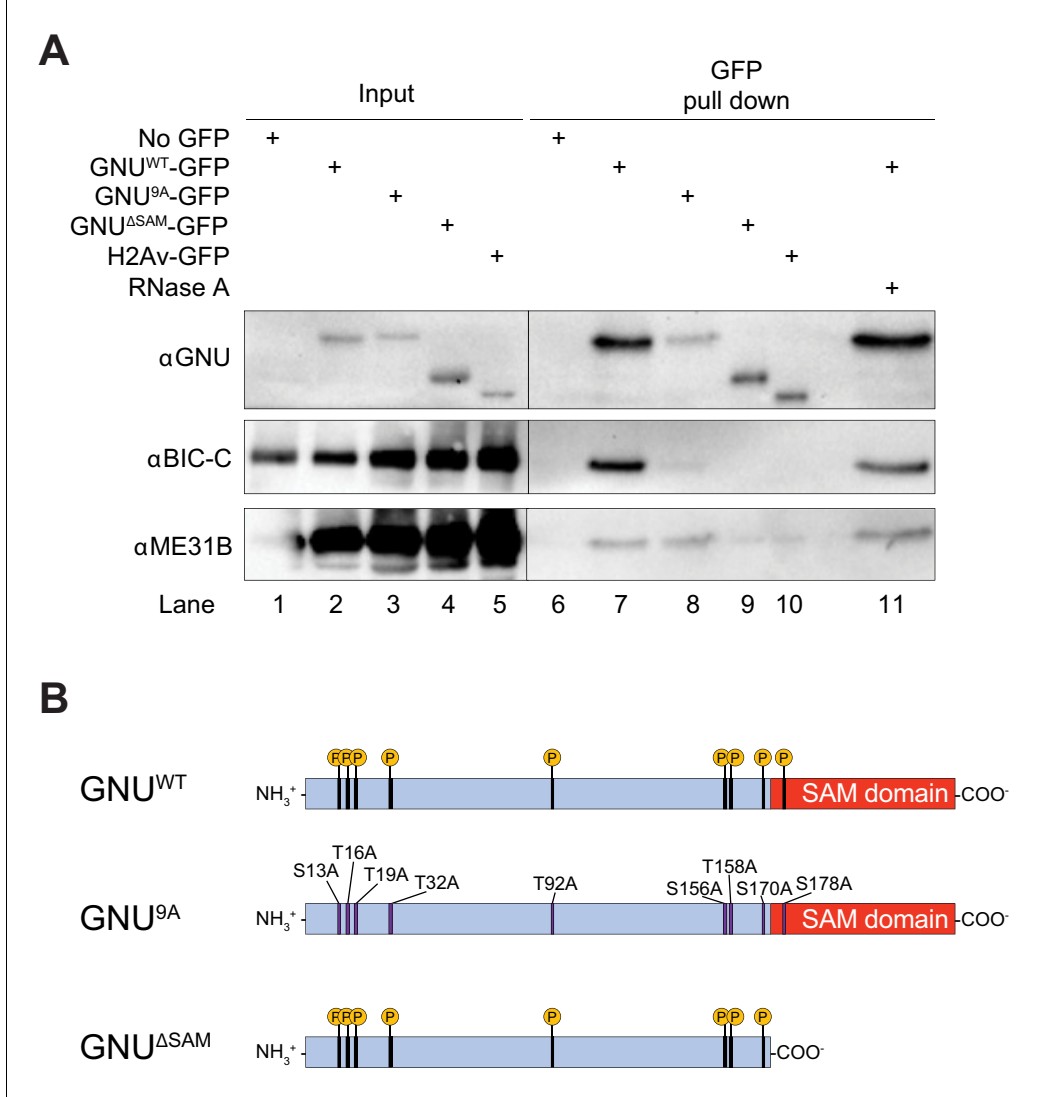

**Figure 1.** GNU physically associates with BIC-C and ME31B in mature oocytes. (**A**) Immunoblot analysis of GNU-GFP immunoprecipitation of extracts from mature oocytes. Anti-GFP magnetic beads were used to perform pull-downs of GNU-GFP from extracts prepared from isolated mature oocytes expressing *gnu^wt^-gfp, gnu^9A^-gfp,* or *gnu^ΔSAM^-gfp* transgenes. GFP immunoprecipitations from no transgene (no GFP) and *h2av-gfp* control extracts controlled for interactions with the beads or GFP tag. GNU^WT^-GFP pull-down results in immunoprecipitation of BIC-C and ME31B (lane 7). Neither BIC-C nor ME31B is immunoprecipitated in no transgene controls (lane 6), but some ME31B is immunoprecipitated with H2Av-GFP (lane 10). Both ME31B and BIC-C are immunoprecipitated by GNU^9A^-GFP (lane 8), whereas some ME31B, but not BIC-C, is immunoprecipitated by GNU^ΔSAM^-GFP (lane 9). Treatment of extracts with 100 µM RNase A does not affect immunoprecipitation of BIC-C or ME31B by GNU^WT^-GFP (lane 11). Quantitation of the immunoblot can be found in *Figure 1—figure supplement 3*. (**B**) Schematic of GNU mutant proteins. The hypophosphorylated mutant GNU, GNU^9A^, has alanine substitutions at all CDK1 phosphorylation sites. The SAM domain of GNU (amino acids 172–240) has been deleted in the GNU^ΔSAM^ mutant protein, while the CDK1 phosphorylation sites remain unaffected. The fusion of GFP to the C-terminus used in the experiments is not shown.

The online version of this article includes the following source data and figure supplement(s) for figure 1:

**Source data 1.** Raw immunoblots from *Figure 1A* and figure with labeled bands.

**Figure supplement 1.** GNU interacts with BIC-C and PP2A subunits.

**Figure supplement 1—source data 1.** Total spectrum counts resulting from proteins identified through IP-MS of GNU-WT-GFP and no GFP control.

**Figure supplement 2.** Specificity of GNU interactions and effect of the CDK1 phosphorylation mutants and deletion of the SAM domain of GNU.

*Figure 1 continued on next page*

*Figure 1 continued*

**Figure supplement 2—source data 1.** Total spectrum counts for GNU-WT-GFP, GNU-deltaSAM-GFP, and H2Av-GFP pull-downs.
**Figure supplement 3.** Quantitation of immunoprecipitated BIC-C bands in *Figure 1A*.
**Figure supplement 3—source data 1.** Quantification of immunoblots shown in *Figure 1A*.

---

hybrid tests (*Chicoine et al., 2007*; *Giot et al., 2003*). We recombinantly expressed and purified MBP-GNU and GST-BIC-C, and performed pull-downs with glutathione sepharose beads. We observed that MBP-GNU was pulled down with GST-BIC-C (*Figure 2A*). Moreover, deleting the SAM domain of GNU resulted in only background levels of GNU in eluates following GST-BIC-C pull-down (*Figure 2A*). Our in vitro experiments support the in vivo results of the requirement of the SAM domain of GNU for BIC-C association and indicate that GNU directly interacts with BIC-C. Given BIC-C also contains a SAM domain (*Gamberi and Lasko, 2012*), it is possible that the interaction between GNU and BIC-C requires the SAM domain of BIC-C as well.

Because the SAM domain of GNU is required for the interaction with BIC-C, we investigated the activity of GNU lacking the SAM domain by analyzing mutant phenotypes. Null mutations in *gnu* result in a characteristic giant nuclei phenotype in embryos due to DNA replication without nuclear division (*Freeman and Glover, 1987*). Rescue of the *gnu* null phenotype is a way to test *gnu* function, so we collected embryos laid by females carrying the *gnu^{ΔSAM}-gfp* transgene to examine the ability of the GNU^{ΔSAM}-GFP protein to rescue a *gnu^{305}* null phenotype.

Two different *gnu^{ΔSAM}-gfp* transgenic lines (3–2 and 3–8), and two different *gnu^{wt}-gfp* transgenic lines (1–5 and 1–8) were analyzed. We also collected embryos from a wild-type strain, as well as from homozygous females for the *gnu^{305}* null allele. We observed only partial rescue of the *gnu* phenotype in embryos laid by *gnu^{ΔSAM}-gfp* females (*Figure 2B,C*). In contrast, most *gnu^{wt}-gfp* developed normally (*Figure 2B,C*). The rescue observed in *gnu^{ΔSAM}-gfp* embryos had also low penetrance; a fraction of embryos developed like a wild-type control, but most showed one or more giant nuclei or defective early divisions. The partial rescue conferred by GNU^{ΔSAM}-GFP is consistent with some GNU functions in this form of the protein.

The giant nuclei phenotype observed in *gnu* mutants is due to a lack of *cycB* translation, which results in DNA replication without nuclear division (*Vardy and Orr-Weaver, 2007*). Translation of *cycB* mRNA upon egg activation is completely dependent on PNG (*Kronja et al., 2014a*; *Lee et al., 2001*; *Vardy and Orr-Weaver, 2007*). Therefore, the levels of CYCB in early embryos can be used to assess PNG activation. Because of our finding that *gnu^{ΔSAM}-gfp* confers partial *gnu* function, we examined the activation of PNG in these mutants by immunoblot analysis of the levels of CYCB in oocytes and early embryos. We found that CYCB was present in early embryos from *gnu^{ΔSAM}-gfp* females, though at lower levels than in *gnu^{wt}-gfp* control embryos (*Figure 2D*, compare lanes 8 and 9 to 10 and 11). The observed decrease in embryonic CYCB levels is consistent with the observed partial rescue of the *gnu* phenotype in *gnu^{ΔSAM}-gfp* embryos.

As an independent confirmation of the ability of GNU^{ΔSAM} to activate PNG, we performed in vitro kinase assays with PNG. In this assay, recombinant PNG kinase complex was incubated with either purified MBP-GNU^{WT} or MBP-GNU^{ΔSAM}, and PNG kinase activity was measured by radiolabeling of Myelin Basic Protein by $\gamma^{32}$P-ATP. We found that incubation with MBP-GNU^{ΔSAM} resulted in PNG kinase activity comparable to incubation with MBP-GNU^{WT} (*Figure 2E*). The fact that *gnu^{ΔSAM}-gfp* only confers partial rescue of *gnu* function in vivo despite the ability of GNU^{ΔSAM}-GFP to fully activate PNG in vitro indicates a requirement of the SAM domain of GNU in the activity of PNG in vivo, perhaps by defining targets for PNG phosphorylation. Alternatively, detection of in vitro kinase activity may be a less sensitive test of PNG function than the phenotypic analysis.

In our immunoblot analysis comparing mature oocytes and early embryos, we also analyzed protein levels of BIC-C, with unexpected results. In oocytes, we observed no difference in BIC-C levels between *gnu^{ΔSAM}-gfp*, *gnu^{305}*, or *gnu^{wt}-gfp* (*Figure 2D*, lanes 2–6). However, we saw decreased BIC-C levels in embryos from both *gnu^{ΔSAM}-gfp* lines compared to embryos from *gnu^{wt}-gfp* (*Figure 2D*, compare lanes 8 and 9 to 10 and 11). Interestingly, this decrease in BIC-C levels in *gnu^{ΔSAM}-gfp* embryos also occurred in *gnu^{305}* embryos (*Figure 2D*, lane 12), showing that lack of the SAM domain is comparable to complete loss of GNU in relation to BIC-C levels. In contrast, no

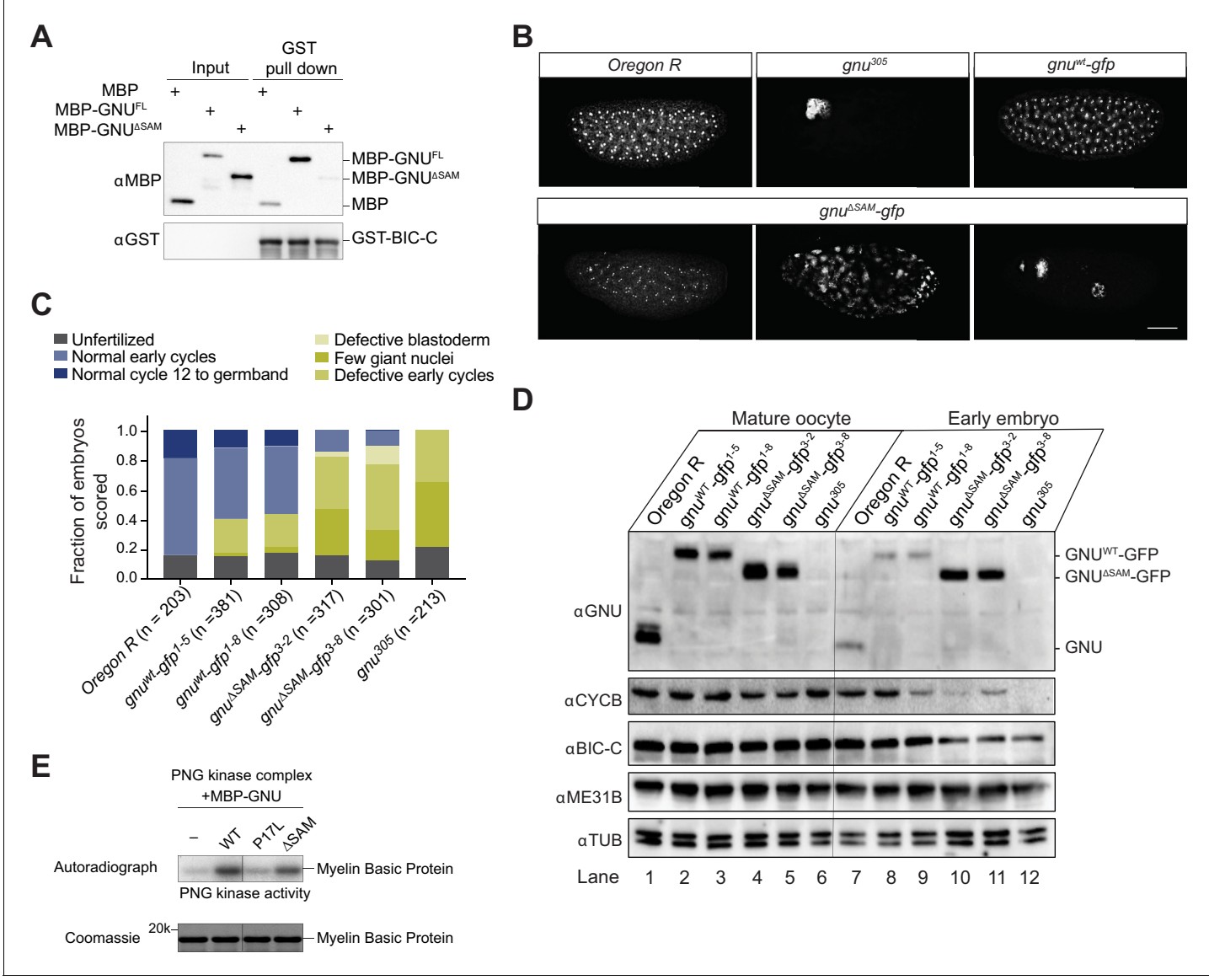

**Figure 2.** Deletion of the SAM domain of GNU reduces the interaction with BIC-C and confers partial GNU function. (**A**) Immunoblot analysis of in vitro pull-down of GNU by BIC-C. Recombinantly expressed and purified MBP-tagged full-length GNU or GNU$^{\Delta SAM}$ was incubated with GST-BIC-C followed by a GST pull-down. As a control, GST pull-down after incubation with MBP was performed. In contrast to the robust pull-down of MBP-GNU, only slight amounts of MBP-GNU$^{\Delta SAM}$ were pulled down by GST-BIC-C. The levels of GST-BIC-C pulled down are comparable between all samples. (**B, C**) Fertilized embryos were collected for 2 hr from wild-type (*Oregon R*), *gnu$^{wt}$-gfp* (*gnu$^{wt}$-gfp; gnu$^{305}$/ gnu$^{305}$*), *gnu$^{\Delta SAM}$-gfp* (*gnu$^{\Delta SAM}$-gfp; gnu$^{305}$/gnu$^{305}$*), or *gnu$^{305}$* (*gnu$^{305}$/gnu$^{305}$*) females. Embryos were fixed and stained with DAPI. (**B**) Representative images of embryonic phenotypes. The embryos from wild-type and *gnu$^{wt}$-gfp* mothers show normal early nuclear division cycles, whereas the *gnu$^{305}$* embryo shows only a few giant nuclei. These nuclei are the consequence of DNA replication in the absence of nuclear division; the number of separate nuclei depends on whether polyploid polar bodies fuse and whether any mitotic divisions occur (***Freeman and Glover, 1987***; ***Lee et al., 2003***). The *gnu$^{\Delta SAM}$-gfp* embryos show (from left to right panels) normal early cycles, defective blastoderm, and a few giant nuclei. Scale bar represents 100 μm. (**C**) Quantification of the embryonic phenotypes. Two independent transgenic lines were analyzed for *gnu$^{wt}$-gfp* (*gnu$^{wt}$-gfp$^{1-5}$* and *gnu$^{wt}$-gfp$^{1-8}$*) and *gnu$^{\Delta SAM}$-gfp* (*gnu$^{\Delta SAM}$-gfp$^{3-2}$* and *gnu$^{\Delta SAM}$-gfp$^{3-8}$*). At least 300 embryos were scored for each transgenic line and at least 200 for the *Oregon R* and *gnu$^{305}$* controls. (**D**) Immunoblot analysis of protein levels in mature oocytes and embryos from *gnu* mutants. Extracts were made from mature oocytes and embryos collected for 1 hr from *Oregon R*, *gnu$^{wt}$-gfp*, *gnu$^{\Delta SAM}$-gfp*, and *gnu$^{305}$* females, and the levels of GNU, CYCB, BIC-C, and ME31B were examined by immunoblot. αTUB was used as a loading control. Two independent transgenic *gnu$^{wt}$-gfp* and *gnu$^{\Delta SAM}$-gfp* lines were examined. 30 oocytes or embryos were collected for each sample, and the equivalent of 10 oocytes was loaded into the gel per sample. Shown is one of two biological replicates. (**E**) In vitro assay of PNG kinase activity. Purified MBP-tagged GNU$^{WT}$, GNU$^{\Delta SAM}$, or GNU$^{P17L}$ was incubated with the recombinant PNG kinase complex and Myelin Basic Protein (an in vitro phosphorylation target of PNG). Levels of phosphorylation of Myelin Basic Protein by PNG with radiolabeled phosphate were measured by autoradiography. MBP-GNU$^{P17L}$ was used as a negative control, as this amino acid change affects the ability of GNU to activate PNG kinase. In

*Figure 2 continued on next page*

*Figure 2 continued*

contrast, both GNU[WT] and GNU[ΔSAM] activate PNG kinase. The levels of Myelin Basic Protein are comparable across samples, as assessed by Coomassie staining.

The online version of this article includes the following source data for figure 2:

**Source data 1.** Raw immunoblots from *Figure 2A* and figure with labeled bands.
**Source data 2.** Total embryo counts for *gnu* rescue experiment results in *Figure 2C*.
**Source data 3.** Raw immunoblots from *Figure 2D* and figure with labeled bands.
**Source data 4.** Raw Coomassie-stained gel and autoradiograph from *Figure 2E*.

---

effect was observed on ME31B protein levels. The mechanism for the decrease in BIC-C in embryos from *gnu*[ΔSAM]*-gfp* mothers is unknown at this time. Previous work demonstrated that PNG function is not required for translation of *Bic-C* mRNA in oocytes or following egg activation (*Kronja et al., 2014a*). As BIC-C can be phosphorylated by PNG (*Hara et al., 2018*), it is possible that the loss of the GNU interaction with BIC-C or the lower level of PNG activity in these mutants reduces phosphorylation of BIC-C by PNG. Consistent with PNG kinase activity being important, quantitative proteomic studies showed that in *png* mutant-activated eggs BIC-C protein levels were reduced relative to wild type (*Kronja et al., 2014b*). This proposal would require that PNG phosphorylation stabilizes BIC-C in contrast to its destabilizing GNU, ME31B, and other proteins (*Wang et al., 2017*).

GNU is degraded in early embryos by a PNG-dependent mechanism (*Hara et al., 2017*). Surprisingly, in our immunoblot analysis of *gnu*[ΔSAM]*-gfp* oocytes and embryos, there were high levels of GNU in *gnu*[ΔSAM]*-gfp* embryos (*Figure 2D*, lanes 10 and 11). In contrast, we observed a decrease in GNU levels in wild-type and *gnu*[wt]*-gfp* embryos (*Figure 2D*, lanes 7–9), consistent with the degradation of GNU following egg activation. The persistence of GNU protein in the *gnu*[ΔSAM]*-gfp* transgene lines suggests that the SAM domain of GNU is necessary for its degradation following egg activation. It is possible that the SAM domain of GNU contains the sites of phosphorylation by PNG necessary for GNU degradation or the recognition site required for proteasome-dependent degradation.

## GNU localizes to cytoplasmic RNP granules in mature oocytes

The finding that GNU interacts with BIC-C and other RNP components prompted us to investigate whether GNU could be part of RNP granules. *Drosophila* oocyte cytoplasmic RNP granules are not restricted to the posterior pole plasm and thus differ from polar granules, being more akin to the P-bodies found in many somatic cell types (*Kato and Nakamura, 2012*). Given that in *Drosophila* oocytes ME31B and TRAL localize to RNP granules (*Nakamura et al., 2001*; *Wilhelm et al., 2005*), the physical association of GNU with ME31B suggests that GNU too may be in at least a subset of these P-body-like cytoplasmic RNP granules.

We examined GNU localization in mature oocytes using the GFP-tagged GNU from *gnu*[wt]*-gfp* transgenic flies. We found that GNU-GFP localized to granular cytoplasmic structures in mature oocytes (*Figure 3B*, top row), whereas no such structures were observed in the absence of a *gnu-gfp* transgene (*Figure 3A*). Moreover, these granular structures were observed in the cytoplasm of oocytes from both *gnu*[wt]*-gfp* transgenic lines (*Figure 3B*, top row). RNP granules have been observed to disperse following egg activation (*Noble et al., 2008*; *Weil et al., 2012*). We therefore also examined the localization of GNU[WT]-GFP following egg activation. We performed in vitro egg activation of *gnu*[wt]*-gfp* oocytes and imaged for GFP fluorescence. We found that GNU[WT]-GFP signal became dispersed through the cytoplasm in in vitro activated eggs from both *gnu*[wt]*-gfp* transgenic lines (*Figure 3B*, bottom row), consistent with GNU granules dissociating upon egg activation. The dispersal of GNU following egg activation is not dependent on PNG function, as we did not observe a difference in localization of GNU[WT]-GFP in *png* mutants (*Figure 3—figure supplement 1B*). Interestingly, PNG and PLU do not exhibit a granular localization (*Figure 3—figure supplement 1C*). This is consistent with GNU not being in a complex with PNG and PLU in mature oocytes and suggests that the observed localization change of GNU might reflect a change in its association with PNG.

To test further whether the particulate pattern of GNU localization in mature oocytes could be because of its presence in cytoplasmic RNPs, we directly compared GNU localization with that of ME31B and TRAL. We examined co-localization in mature oocytes between TRAL-GFP and ME31B-

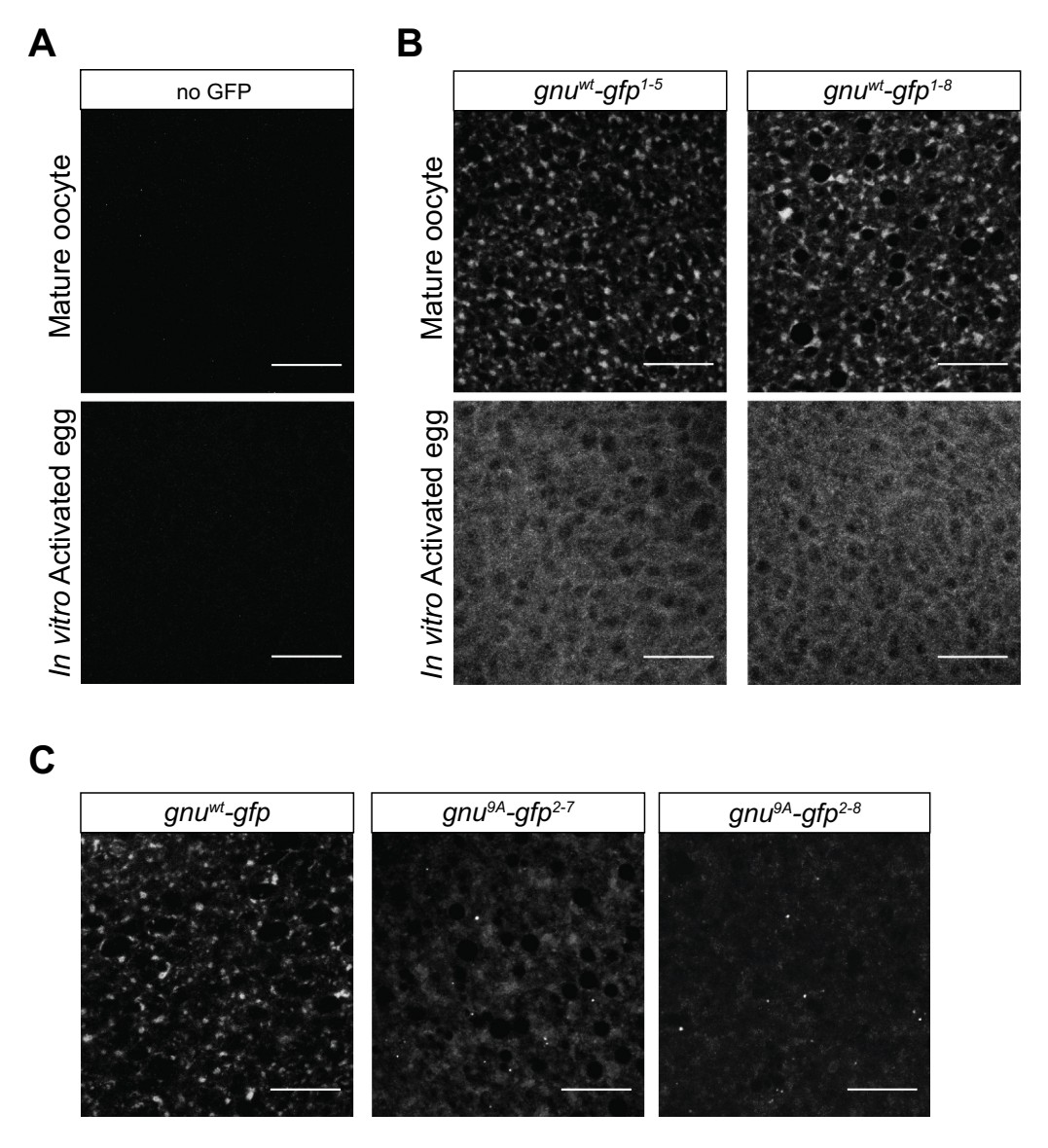

**Figure 3.** GNU localizes to granular cytoplasmic structures in mature oocytes. Mature oocytes were isolated from *gnu^wt-gfp* or *gnu^9A-gfp* transgenic females. Oocytes were fixed and the vitelline membrane was removed manually before staining with the anti-GFP booster. A no GFP transgene control was analyzed for comparison. For imaging of activated eggs, mature oocytes were isolated and activated in vitro by incubation in the hypotonic buffer for 20 min. Successfully activated eggs were selected by bleach treatment, fixed with methanol, and stained with the anti-GFP booster and Hoechst 33342. Each image is a maximum intensity projection from five stacks of a z-series of the cytoplasm of one mature oocyte or activated egg. Scale bars represent 20 μm. (**A**) Representative images of no GFP transgene oocyte (top panel) and activated egg (bottom panel), stained with an anti-GFP booster. (**B**) Representative images of *gnu^wt-gfp* transgenic oocyte (top panels) and activated egg (bottom panels), stained with an anti-GFP booster. Two different *gnu^wt-gfp* transgenic lines (*gnu^wt-gfp^1-5* and *gnu^wt-gfp^1-8*) were analyzed. A granular cytoplasmic localization pattern is observed for GNU^WT-GFP. (**C**) Representative images of *gnu^9A-gfp* transgenic oocytes, stained with an anti-GFP booster. Two different *gnu^9A-gfp* transgenic lines (*gnu^9A-gfp^2-7* and *gnu^9A-gfp^2-8*) were analyzed. A representative image of *gnu^wt-gfp* oocytes is shown for comparison. A diffuse cytoplasmic signal and bright puncta are observed for GNU^9A-GFP in oocytes from both *gnu^9A-gfp* lines.

The online version of this article includes the following source data and figure supplement(s) for figure 3:

**Figure supplement 1.** Comparison of GFP-tagged GNU levels in *gnu-gfp* transgenic lines, localization of PLU and PNG in mature oocytes, and GNU-GFP localization in *png* mutant oocytes.

**Figure supplement 1—source data 1.** Raw immunoblots from *Figure 3—figure supplement 1* and figure with labeled bands.

GFP, and GNU^(WT)-mKATE2. In these experiments, GNU^(WT)-mKATE2 is expressed in the presence of endogenous GNU, though GNU^(WT)-mKATE recapitulated the localization described for GNU^(WT)-GFP (*Figure 4—figure supplement 1A,A'*). No significant co-localization was observed between GNU^(WT)-mKATE2 and the ER marker, PDI-GFP (*Figure 4—figure supplement 1B,B'*). We found that GNU^(WT)-mKATE2 co-localized with both TRAL-GFP and ME31B-GFP in cytoplasmic granules (*Figure 4*). Whereas most GNU^(WT)-mKATE2 co-localized with TRAL-GFP and ME31B-GFP, only a fraction of all observed TRAL-GFP and ME31B-GFP granules also contained GNU^(WT)-mKATE2 (*Figure 4A', B'*). These observations are consistent with GNU localizing to RNP granules, with GNU-containing granules representing a subpopulation of TRAL- and ME31B-containing RNP granules present in mature oocytes.

The observed change in GNU localization in activated eggs suggests potential regulation of GNU localization by its CDK1 phosphorylation state, as GNU becomes hypo-phosphorylated at egg activation (*Hara et al., 2017*). The localization of GNU^(9A)-GFP in mature oocytes from two different *gnu^(9A)-gfp* (2–7 and 2–8) transgene lines was visualized. We observed mostly dispersed localization for GNU^(9A)-GFP in the oocyte cytoplasm, although there were some bright cytoplasmic puncta present (*Figure 3C*). The localization of GNU^(9A)-GFP contrasts with the granular localization observed for GNU^(WT)-GFP. The expression levels of GNU^(9A)-GFP and GNU^(WT)-GFP were comparable between *gnu^(wt)-gfp* and *gnu^(9A)-gfp* transgene lines (*Figure 3—figure supplement 1A*), so the differences were not due to altered GNU levels. The difference in localization of GNU^(9A)-GFP as compared to wild type is consistent with a role for CDK1 phosphorylation in modulating GNU localization. In additional, as GNU^(9A)-GFP can still bind some RNP components (albeit with reduced affinity for BIC-C, see above), the binding to those RNP components appears insufficient for the localization of GNU to RNP granules, and CDK1 phosphorylation may also be necessary.

## GNU is dependent on BIC-C for cytoplasmic RNP localization

During oogenesis, BIC-C forms a complex with both TRAL and ME31B, and it is also a component of cytoplasmic RNP granules (*Kugler et al., 2009*). To test whether BIC-C and GNU are present in the same RNP granules, we examined the co-localization of GNU^(WT)-mKATE2 with BIC-C-GFP in mature oocytes. GNU^(WT)-mKATE2 and BIC-C-GFP co-localized (*Figure 5A*). Moreover, whereas most GNU granules contained BIC-C, GNU-containing granules represented only a fraction of all BIC-C granules (*Figure 5A'*). This observation is consistent with GNU being part of a subpopulation of RNP granules in mature oocytes. Because BIC-C and GNU co-localize and the interaction between these proteins depends on the SAM domain of GNU, we also investigated a role for the SAM domain in GNU localization. We found that in contrast to the granular localization observed in GNU^(WT)-GFP, GNU^(ΔSAM)-GFP exhibited a dispersed localization in the cytoplasm of mature oocytes (*Figure 5B*), evidence that the SAM domain of GNU is required for GNU localization to RNP granules.

The co-localization of BIC-C and GNU, and the requirement of the SAM domain of GNU suggest that BIC-C might be required for GNU localization. We thus tested the effect of reduction of functional BIC-C on GNU localization in mature oocytes by looking at GNU^(WT)-GFP localization in *Bic-C* mutant mature oocytes. Mutations in *Bic-C* that significantly reduce or abolish *Bic-C* function result in a dominant embryonic patterning defect (*Schüpbach and Wieschaus, 1991*), as well as in an arrest in early oogenesis in homozygous mutant females (*Mahone et al., 1995*). We used the *Bic-C^4*/ *Bic-C^(PE37)* allele combination to significantly reduce BIC-C levels in mature oocytes (*Figure 5—figure supplement 1A*). The *Bic-C^4* allele results in no BIC-C protein, and the *Bic-C^(PE37)* allele reduces protein levels (*Saffman et al., 1998*). Female flies carrying this combination of alleles have a reduction of *Bic-C* function during oogenesis but still produce mature oocytes. We examined GNU^(WT)-GFP localization in *Bic-C^4/Bic-C^(PE37)* mature oocytes, as well as in heterozygotes for each allele (*Bic-C^4/+* and *Bic-C^(PE37)/+*). We observed no visible differences in GNU^(WT)-GFP in heterozygous *Bic-C* mutants (*Figure 5C*). However, GNU^(WT)-GFP localization was disrupted in *Bic-C^4/Bic-C^(PE37)* mutant oocytes. GNU-GFP exhibited a mostly dispersed appearance in these oocytes, reminiscent of GNU^(ΔSAM)-GFP localization, but with a portion of GNU-GFP still localized to granules. The effect of *Bic-C* mutations on GNU^(WT)-GFP localization raises the possibility that the reduced granular localization observed for GNU^(9A)-GFP (*Figure 3C*) could be due in part to its weakened interaction with BIC-C (*Figure 1A*, *Figure 1—figure supplement 2B*). Notably, the disruption of GNU localization in the *Bic-C* mutant is less severe than the localization of GNU^(9A)-GFP. The effect of mutations in *Bic-C* on GNU^(WT)-GFP was not due to differences in GNU protein levels, as immunoblot analysis (*Figure 5—figure*

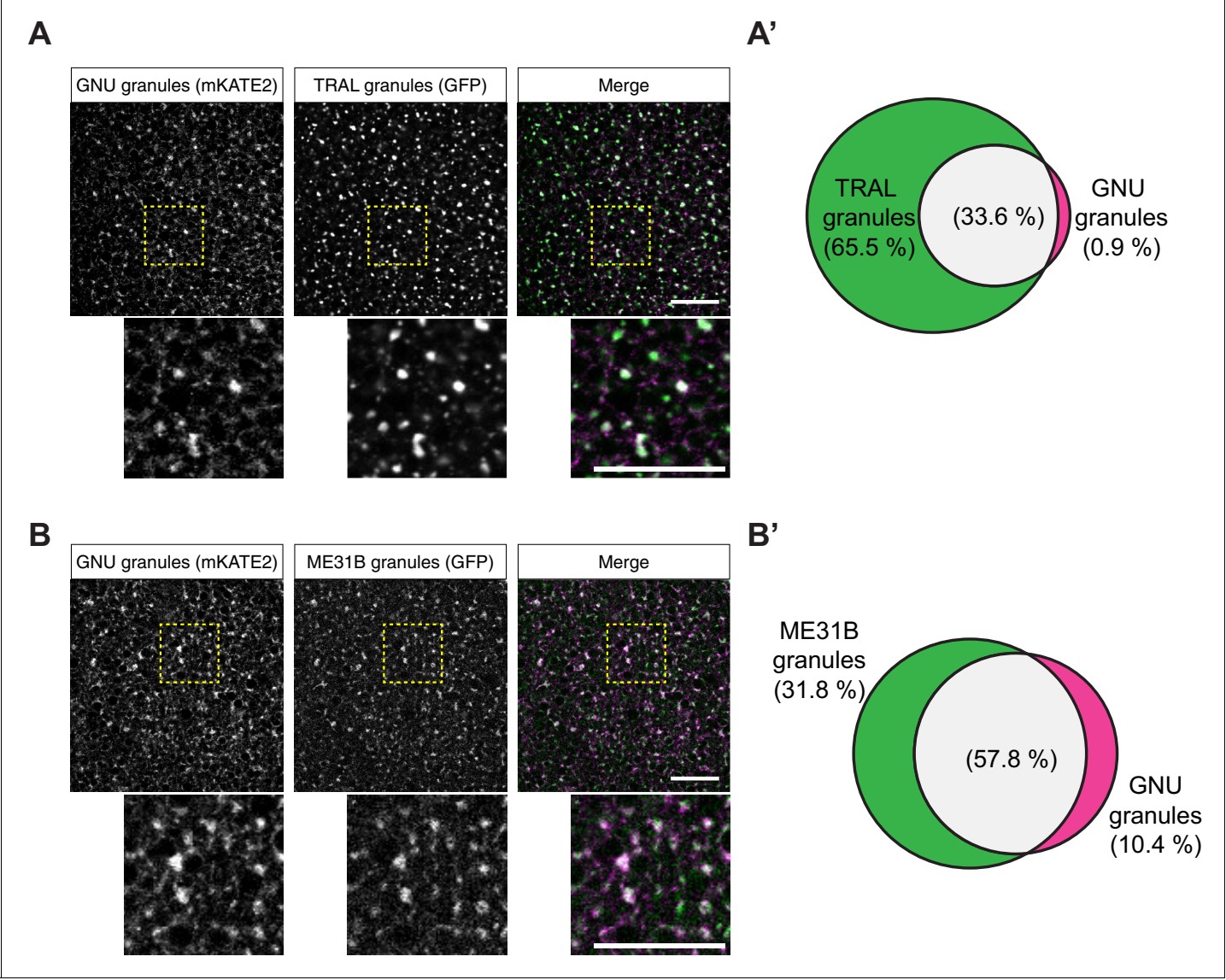

**Figure 4.** GNU-mKATE2 co-localizes with TRAL-GFP and ME31B-GFP granules in mature oocytes. Mature oocytes were isolated from *gnu$^{wt}$-mkate2; tral-gfp* or *me31b-gfp;gnu$^{wt}$-mkate2* females, fixed, and the vitelline membrane removed manually. Oocytes were stained with the anti-GFP booster and imaged by confocal microscopy for fluorescence at 488 nm to detect GFP and 568 nm to detect mKATE2. mKATE2 signal was detected without the use of a booster. Co-localization was measured by quantification of overlap between GFP+ granules and mKATE2+ granules using the surface-surface co-localization algorithm in Imaris (Bitplane). (**A**) Representative image of *gnu$^{wt}$-mkate2;tral-gfp* oocytes. Co-localizing GNU-mKATE2 (magenta) and TRAL-GFP (green) granules are colored in white. The images shown are single slices of confocal z-stacks from one oocyte. Bottom images show the insets of each panel (dashed yellow box). Scale bar represents 20 μm. (**A'**) Venn diagram of quantified co-localization between GNU and TRAL granules. GNU and TRAL co-localize in 33.6±5.2% of all granules quantified. GNU-containing TRAL granules represent approximately a third of TRAL granules scored. Values are averaged across eight oocytes. (**B**) Representative image of *me31b-gfp;gnu$^{wt}$-mkate2* oocytes. Co-localizing GNU-mKATE2 (magenta) and ME31B-GFP (green) granules are colored in white. The images shown are single slices of confocal z-stack from one oocyte. Bottom images show the insets of each panel (dashed yellow box). Scale bar represents 20 μm. (**B'**) Venn diagram of quantified co-localization between GNU and ME31B granules. GNU and ME31B co-localize in 57.8±4.6% of all granules quantified. GNU-containing ME31B granules represent approximately half of ME31B granules scored. Values are averaged across eight oocytes.

The online version of this article includes the following source data and figure supplement(s) for figure 4:

**Source data 1.** Quantification data for co-localization experiments in *Figure 4* and *Figure 4—figure supplement 1*.

**Figure supplement 1.** GNU-mKATE recapitulates mature oocyte localization of GNU-GFP.

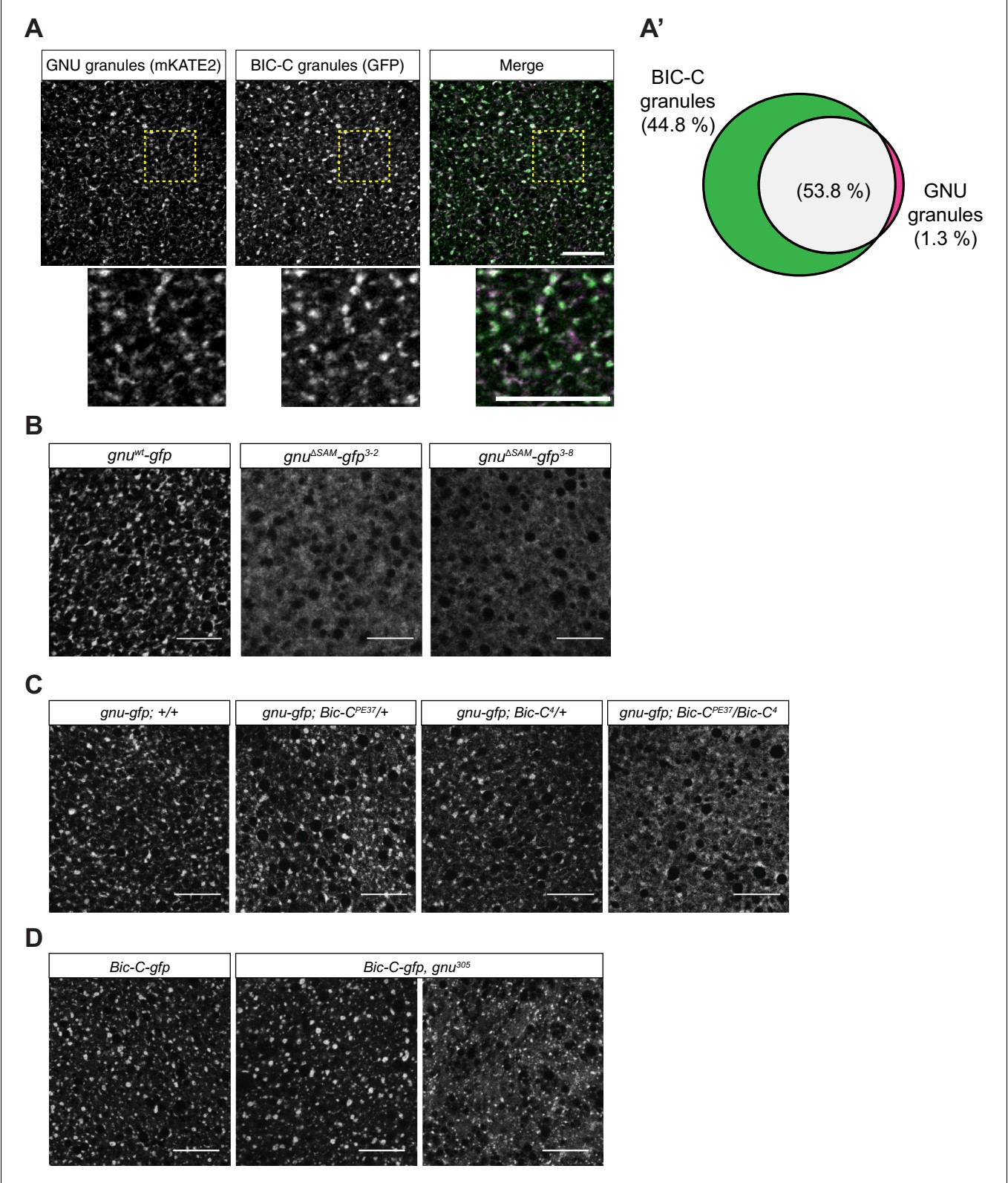

**Figure 5.** GNU localization to granules is dependent on its SAM domain, and BIC-C and GNU are co-dependent for localization. (**A, A'**) Co-localization between GNU-mKATE2 and BIC-C-GFP. Mature oocytes were isolated from *gnu^wt^-mkate2;Bic-C-gfp* females and imaged and analyzed as in *Figure 4*. (**A**) Representative image of *gnu^wt^-mkate2;Bic-C-gfp* oocyte. Co-localizing GNU (magenta) and BIC-C (green) granules are colored in white. Bottom images show the insets of each panel (dashed yellow box). (**A'**) Venn diagram of quantified co-localization between GNU and BIC-C granules. GNU and

*Figure 5 continued on next page*

*Figure 5 continued*

BIC-C co-localize in 53.8±4.8% of all granules quantified. GNU-containing BIC-C granules represent approximately half of all BIC-C granules scored. Values are averaged across eight oocytes. (B) Representative images of $gnu^{\Delta SAM}$-*gfp* transgenic oocytes, stained with an anti-GFP booster. Two different $gnu^{\Delta SAM}$-*gfp* transgenic lines ($gnu^{\Delta SAM}$-*gfp*$^{3-2}$ and $gnu^{\Delta SAM}$-*gfp*$^{3-8}$) were analyzed. A representative image of $gnu^{wt}$-*gfp* oocytes is shown for comparison. A diffuse cytoplasmic localization is observed for GNU$^{\Delta SAM}$-GFP in oocytes from both $gnu^{\Delta SAM}$-*gfp* transgenic lines. (C) Representative images of GNU$^{WT}$-GFP localization in *Bic-C* mutant mature oocytes. Localization of GNU$^{WT}$-GFP is comparable between $gnu^{wt}$-*gfp; +/+*, and heterozygous oocytes for *Bic-C* loss-of-function alleles ($gnu^{wt}$-*gfp; Bic-C*$^{PE37}$/+ and $gnu^{wt}$-*gfp; Bic-C*$^4$/+). GNU$^{WT}$-GFP localization is more diffuse in $gnu^{wt}$-*gfp; Bic-C*$^{PE37}$/*Bic-C*$^4$ mutants. (D) Representative images of BIC-C-GFP localization in *gnu* mutant mature oocytes. BIC-C-GFP localizes to granules in control *Bic-C-gfp* oocytes. Localization of BIC-C-GFP in *Bic-C-gfp;gnu*$^{305}$ looks comparable to *Bic-C-gfp* control in 70% of oocytes scored (n=20), with 30% of scored oocytes exhibiting a dispersed localization with punctate granules for BIC-C-GFP. The *gnu*$^{305}$ allele is a protein null allele of *gnu* (*Renault et al., 2003*). In (A–D), scale bars represent 20 µm. In (B–D), the images shown are a maximum intensity projection of a z-series of five stacks from one oocyte, whereas in (A), the images shown are single slices of confocal z-stack from one oocyte.

The online version of this article includes the following source data and figure supplement(s) for figure 5:

**Source data 1.** Quantification data for co-localization experiments shown in *Figure 5A*.
**Figure supplement 1.** BIC-C levels are decreased in *Bic-C* mutant mature oocytes and ME31B localization is not affected in *gnu* mutants.
**Figure supplement 1—source data 1.** Raw immunoblots from *Figure 5—figure supplement 1A* and figure with labeled bands.

*supplement 1A*) yielded no significant difference in GNU levels across these genotypes. We conclude that BIC-C is required for the presence of GNU in RNP granules, either for recruitment or retention, or perhaps both.

To test whether there is a reciprocal dependency and loss of *gnu* function that affects localization of BIC-C in mature oocytes, we examined BIC-C-GFP localization in a *gnu*$^{305}$ null mutant background. We found that loss of *gnu* function resulted in an observable effect on BIC-C-GFP localization (*Figure 5D*), because the localization of BIC-C became partially dispersed in 30% of *gnu* mutant oocytes. We also observed that in oocytes where BIC-C-GFP localization was affected, some BIC-C-GFP granules were still observed. Given that nearly half of BIC-C granules do not contain GNU, it was expected that in the absence of GNU only a subset of the BIC-C RNPs would be affected. BIC-C-GFP localization was unaffected in 70% of oocytes. Our result suggests that the interaction between GNU and BIC-C, while not necessary for BIC-C localization in mature oocytes, might be modulating or stabilizing BIC-C in granules. However, the disruption of BIC-C-GFP in *gnu* mutant oocytes could be due to the disruption of RNP granules in these mutants. To address this issue, we also examined the localization of ME31B-GFP in *gnu* mutant oocytes, finding no effect on ME31B-GFP in these mutants (*Figure 5—figure supplement 1B*). This observation indicates that the effect of *gnu*$^{305}$ on BIC-C-GFP localization is likely due to the loss of the interaction between BIC-C and GNU and not the consequence of a general effect on RNP granules.

## Tests of a potential role for localization in RNP granules as a sequestration mechanism for GNU

GNU does not bind to PNG in mature oocytes, and this appears to be the key way by which activation of PNG kinase is limited until the completion of meiosis. One model for the function of localization of GNU to RNP granules is that it serves to sequester GNU away from PNG until GNU is released from granules at egg activation (Figure 7A). This model makes two predictions. The first is that GNU would no longer be associated with BIC-C after egg activation, because BIC-C function and binding serve to localize GNU to RNPs. The second is that release of GNU from the granules by disruption of the association with BIC-C in the ΔSAM mutant would lead to premature activation of PNG kinase in the mature oocyte, as has been previously observed for the GNU$^{9A}$ mutant (*Hara et al., 2017*). We tested each of these predictions experimentally.

We repeated the immunoprecipitation experiments to test for association between GNU and BIC-C, this time comparing mature oocytes with in vitro activated eggs (*Figure 6A*). GNU clearly retained its association with BIC-C after egg activation, a result not readily consistent with the sequestration model.

Hypo-phosphorylation of GNU in oocytes leads to premature activation of PNG and a significant increase in CYCB levels in mature oocytes (*Hara et al., 2017*). To investigate whether the release of GNU from granules would cause activation of PNG, we collected oocytes from $gnu^{\Delta SAM}$-*gfp* and $gnu^{wt}$-*gfp* flies and measured levels of CYCB by immunoblot analysis (*Figure 6B*). Levels of CYCB

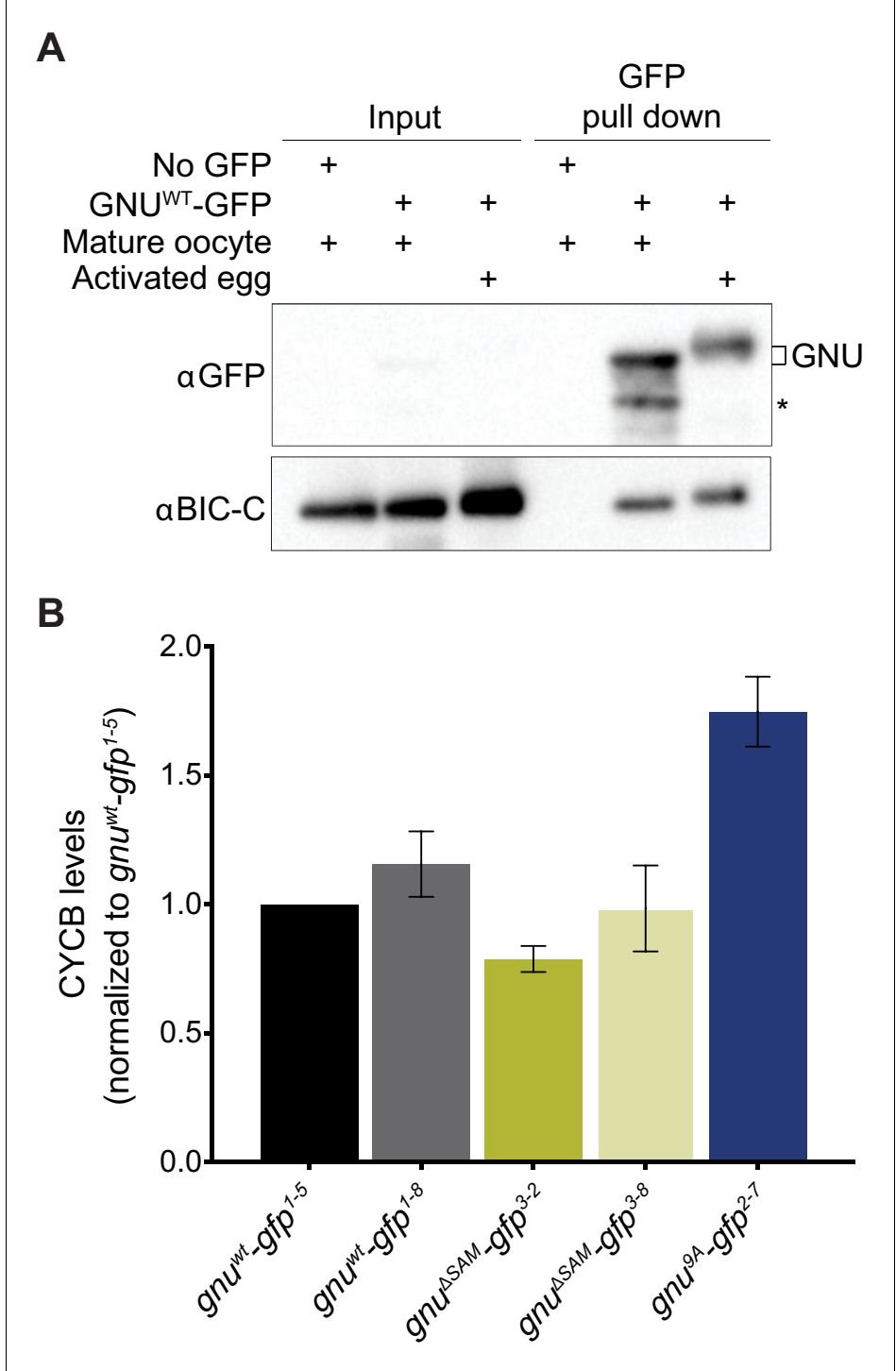

**Figure 6.** Experimental test for model that RNP granule localization of GNU prevents activation of PNG. (**A**) BIC-C and GNU remain physically associated after egg activation. Anti-GFP magnetic beads were used to perform pull-downs of GNU-GFP from extracts prepared from isolated mature oocytes or in vitro activated eggs expressing *gnu^wt^-gfp* transgenes. For the analysis of activated eggs, mature oocytes were isolated and activated in vitro by incubation in hypotonic buffer for 20 min. GFP immunoprecipitations from no transgene (no GFP) mature oocyte extracts controlled for interactions with the beads or GFP tag. GNU-GFP pull-down from both mature oocyte or activated egg extracts results in immunoprecipitation of BIC-C. The asterisk marks a GNU-GFP degradation product we often observe in immunoprecipitations from mature oocytes. (**B**) Deletion of the SAM domain in GNU does not increase levels of CYCB in mature oocytes. Mature oocytes were isolated from *gnu^305^* homozygous

*Figure 6 continued on next page*

*Figure 6 continued*

females expressing *gnu$^{wt}$-gfp, gnu$^{\Delta SAM}$-gfp,* or *gnu$^{9A}$-gfp* transgenes. The levels of CYCB and αTUB were examined by immunoblot. Two independent lines were analyzed for each transgene, except for *gnu$^{9A}$-gfp* for which only one line was analyzed. Levels of CYCB were quantified and normalized to TUB levels. The graph shows normalized levels of CYCB relative to *gnu$^{wt}$-gfp$^{1-5}$* oocytes. Error bars correspond to SEM, and each bar represents five biological replicates. CYCB levels were not significantly different between oocytes from the two *gnu$^{wt}$-gfp* lines (paired t-test, p=0.2855). No significant difference was observed between *gnu$^{wt}$-gfp* and *gnu$^{\Delta SAM}$-gfp$^{3-8}$* (paired t-test, p=0.9281), but the CYCB levels in *gnu$^{\Delta SAM}$-gfp$^{3-2}$* oocyte were significantly lower than in *gnu$^{wt}$-gfp$^{1-5}$* oocytes (paired t-test, *p=0.0137). Levels of CYCB in *gnu$^{9A}$-gfp$^{2-7}$* oocytes are significantly higher compared to *gnu$^{wt}$-gfp* (paired t-test, **p=0.0053).

The online version of this article includes the following source data for figure 6:

**Source data 1.** Quantification of immunoblots shown in *Figure 6A*.
**Source data 2.** Raw immunoblots from *Figure 6A* and figure with labeled bands.
**Source data 3.** Relative levels of CYCB in mature oocytes expressing GNU transgenes.

were comparable or slightly lower in *gnu$^{\Delta SAM}$-gfp* oocytes than in *gnu$^{wt}$-gfp* mature oocytes. There was a small but significant reduction of CYCB levels in *gnu$^{\Delta SAM}$-gfp$^{3-2}$* mature oocytes, but not in *gnu$^{\Delta SAM}$-gfp$^{3-8}$* mature oocytes, as compared to *gnu$^{wt}$-gfp* oocytes (*Figure 6B*). In contrast, as previously reported, we observed a significant increase in CYCB levels in *gnu$^{9A}$-gfp* control oocytes. These results are consistent with the loss of granule localization not being sufficient to activate PNG. However, it is possible that other mRNA targets of regulation by PNG are affected in *gnu$^{\Delta SAM}$-gfp* oocytes.

## Discussion

In this study, we demonstrate that GNU, the lynchpin to the developmentally regulated activation of PNG (*Hara et al., 2017*), is a previously unidentified component of RNP granules in oocytes. We find that GNU forms a complex with translational repressors in mature oocytes. Moreover, not only does GNU interact with the known components of oocyte RNP granules, ME31B and BIC-C, it also co-localizes with these two proteins as well as TRAL in large cytoplasmic structures. In addition to interactions or co-localization with GNU, all three of these proteins are phosphorylated by PNG (*Hara et al., 2018*).

These results reveal the features of GNU required for RNP granule localization: the SAM domain and CDK1 phosphorylation sites. The observation that the SAM domain is required for the localization of GNU to granules as well as its interaction with BIC-C is consistent with a model in which GNU is recruited to RNP granules via specific protein-protein interactions through its SAM domain. Once recruited to these complexes, GNU could indirectly interact with RNAs but without depending on them for its recruitment, which would account for the observation that the protein-protein interactions of GNU are largely unaffected by treatment with RNase. Interestingly, in addition to the SAM domain, GNU also contains an intrinsically disordered region (IDR) that contains eight of the nine CDK1 phosphorylation sites (*Hara et al., 2017*). Studies on RNP granule assembly have suggested that IDRs, although alone not sufficient for granule localization, can form weak non-specific interactions that stabilize their localization within RNP granules (*Lin et al., 2015*; *Protter et al., 2018*). We found that when the SAM domain of GNU is deleted, leaving the IDR intact, the truncated GNU was not able to localize to granules. Thus, while not sufficient for localization in RNP granules, the IDR could stabilize GNU in these complexes.

Posttranslational modifications, such as phosphorylation, can regulate the recruitment of proteins into RNP granules. For example, in mammalian axons, phosphorylation of FMRP promotes assembly of FMRP granules, and in response to action potentials, dephosphorylation of FMRP promotes their disassembly (*Tsang et al., 2019*). The opposite effect of phosphorylation can occur; in *Caenorhabditis elegans* phosphorylation of the MEG granule proteins promotes granule disassembly (*Wang et al., 2014*). Our finding that the CDK1 phosphorylation sites of GNU play a role in localizing GNU suggests potential modulation of GNU RNP recruitment by CDK1 phosphorylation. The observation that GNU$^{9A}$ has reduced granular localization is consistent with the idea that CDK

phosphorylation may act to enhance or stabilize the interaction with RNP granules. Thus, GNU phosphorylation appears to drive, at least in part, its recruitment into granules, as with FMRP. Given the localization of the CDK1 phosphorylation sites of GNU within the IDR domain, CDK1 phosphorylation could act by modifying structural features of the IDR that might affect GNU localization to granules in mature oocytes.

Here, we identified an in vivo interaction between BIC-C and GNU in mature oocytes dependent on the SAM domain of GNU; we propose this is the main mechanism by which GNU is recruited into RNP granules. This interaction had previously only been shown to occur as a two-hybrid interaction (*Chicoine et al., 2007*; *Giot et al., 2003*). The observations that the interaction with BIC-C requires the SAM domain of GNU, but that the interactions with other proteins such as YPS do not, indicates that the interaction of YPS with GNU is unlikely to be mediated through BIC-C. Similarly, the interaction of ME31B with GNU is at least in part independent of BIC-C. In addition, a key role for GNU-BIC-C interactions in the localization of GNU is supported by the dispersion on GNU in *Bic-C* mutant oocytes. It appears that CDK1/CYCB phosphorylation of GNU in mature oocytes aids in stabilizing GNU in RNP granules, possibly through strengthening the interaction of GNU with BIC-C in these structures. Because BIC-C-containing granules are present in oocytes starting at mid-oogenesis (*Kugler et al., 2009*), and GNU levels increase later during oocyte maturation (*Hara et al., 2017*; *Kronja et al., 2014b*), GNU may be recruited onto pre-existing granules as its levels increase. The increasing levels of CDK1/CYCB activity during late oogenesis would further promote the recruitment of GNU into RNP granules.

Our findings also reveal a synergistic relationship between GNU and BIC-C in granules. We found an effect on BIC-C localization in *gnu* mutant oocytes, where BIC-C granules are diminished and take on a more dispersed appearance. The affected granules are presumed to be the subpopulation of BIC-C granules that also contain higher levels of GNU. If so, this suggests that GNU has a role in maintaining or stabilizing BIC-C in granules. We also observed a requirement of the SAM domain of GNU, and likely the interaction with BIC-C, in maintaining BIC-C protein levels in early embryos, which indicates modulation of BIC-C stability by GNU.

An implication of our work is the presence of a diverse pool of RNP granule subtypes in the mature oocyte. Most studies of oocyte RNP granules have been limited to earlier stages of oogenesis, whole-ovary extracts, or have relied on a limited set of markers to observe RNP granules in oocytes. These approaches have precluded the identification of subtypes of RNP granules in mature oocytes, except in the germplasm at the oocyte posterior where germ granules are heterogeneous for constituent mRNAs and form as homotypic clusters by self-recruitment of specific mRNAs (*Niepielko et al., 2018*). A recent study characterizing the interaction of six RNA-binding proteins through GFP-immunoprecipitation of proteins from whole *Drosophila* ovary extracts, followed by mass spectrometry-based protein identification, identified both known and some new interactors with polar granule and nuage granule proteins, but did not identify GNU as an interactor (*Bansal et al., 2020*). In contrast, by examining the interactions of GNU at a specific stage in oogenesis, we found that GNU indeed interacts with several RNA-binding proteins and known RNP granule components. In the current study, we detected GNU only in a subpopulation of TRAL, ME31B, and BIC-C RNP granules in mature oocytes. Because TRAL and ME31B are thought to be present in all granules (*Kato and Nakamura, 2012*), the present findings identify GNU-containing granules as a new subpopulation of oocyte RNP granules. To our knowledge, this is also the first report of the presence of BIC-C granules in mature *Drosophila* oocytes. These different RNP granule subtypes are likely to reflect different roles in the regulation of maternal mRNAs, and they highlight the complexity of the regulation of maternal mRNAs by oocyte RNP complexes.

The demonstration that GNU is a component of RNP granules has led us to rethink how it regulates PNG activity upon egg activation. Our data can be explained by two alternative models. In the first model, the recruitment of GNU onto RNP granules functions as a sequestration mechanism to prevent premature activation of PNG prior to egg activation (*Figure 7A*). In this 'sequestration' model, GNU is prevented from interacting with the PNG/PLU complex by spatial separation from PNG. Upon egg activation, RNP granules disassemble, thus releasing GNU as it is being dephosphorylated. In activated eggs, GNU is no longer prevented from binding PNG by sequestration in RNP granules, and PNG phosphorylates its substrates. The interaction of GNU with PNG targets could also function as a mechanism for substrate recognition. Although much of our data are consistent with this model, both the finding that GNU is still bound to BIC-C after egg activation and the

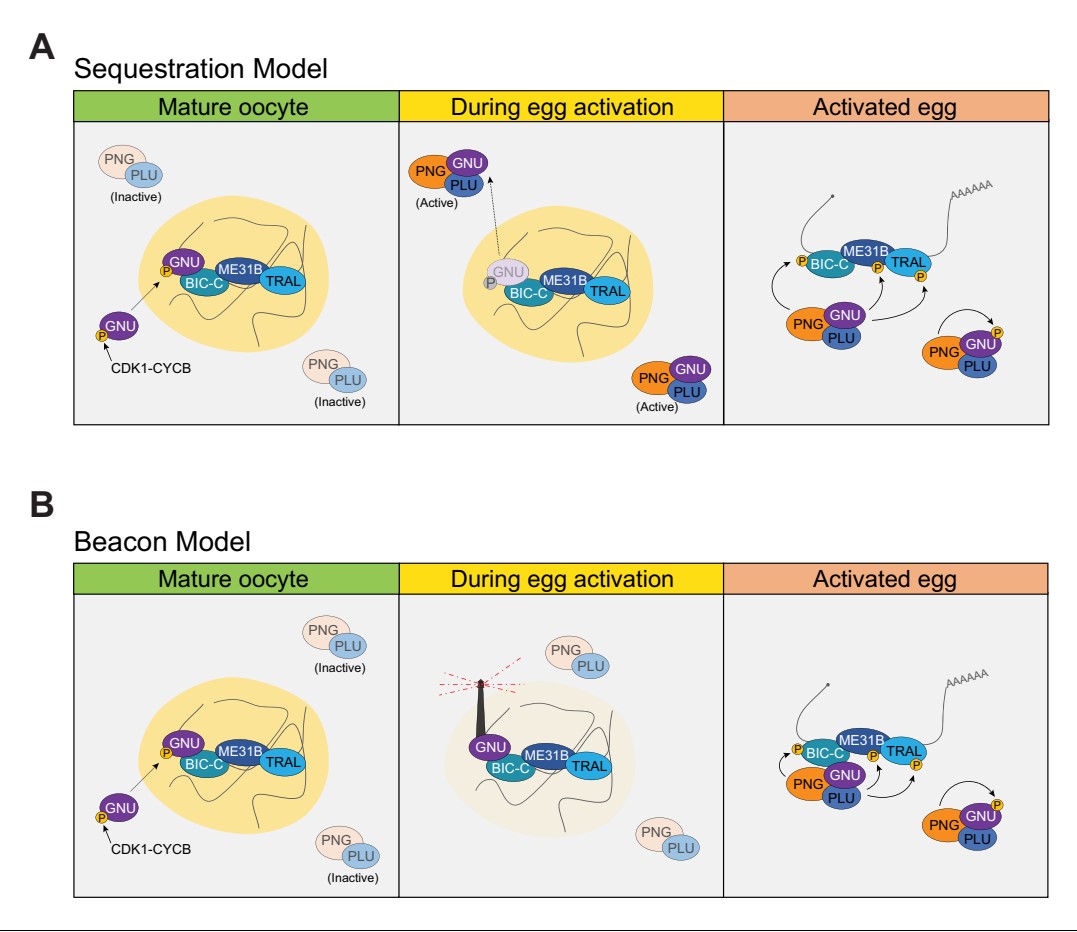

**Figure 7.** Models of regulation of PNG by GNU in RNP granules. (**A**) Sequestration Model. In this model, GNU is prevented from interacting with the PNG/PLU complex by spatial separation. In mature oocytes, GNU is sequestered to granules in a CDK1 phosphorylation and SAM-domain-dependent manner (left panel), where it interacts with BIC-C, ME31B, and TRAL. PNG and PLU are localized throughout the oocyte cytoplasm. Upon egg activation, RNP granules disassemble, thus releasing GNU as it is being dephosphorylated (middle panel). In activated eggs, GNU is no longer prevented from binding PNG by sequestration of RNP granules, and PNG mediates the phosphorylation of its targets as well as autoregulation through phosphorylation of its subunits (right panel). (**B**) Beacon model. In this model, localization of GNU to RNP granules functions to localize PNG activity to these granules. In mature oocytes, GNU is recruited onto granules via SAM domain interactions, with CDK1 phosphorylation stabilizing its interactions within the granules (left panel). Inactive PNG is diffuse throughout the cytoplasm. As egg activation occurs, GNU is dephosphorylated and brings PNG to the disassembling granules, where it can phosphorylate TRAL, ME31B, BIC-C, and potentially other translation regulators (middle panel). In fully activated eggs, PNG activity would not be restricted to granules previously marked by GNU, and the complex is able to phosphorylate targets throughout the activated egg prior to inactivation of the complex (right panel). The beacon model is more consistent with our data than the sequestration model.

observation that loss of RNP granule localization of GNU is not sufficient to prematurely activate PNG argue against the sequestration model.

The data are more consistent with a 'beacon' model of regulation of PNG (*Figure 7B*). In this model, GNU is recruited into granules as a mechanism to mark these granules to target PNG to them. As egg activation occurs, GNU is dephosphorylated and recruits PNG to the disassembling granules, where it can phosphorylate TRAL, ME31B, BIC-C, and potentially other translation regulators. In fully activated eggs in which granules are dissociated, PNG activity would not be spatially restricted, being able to phosphorylate substrates throughout the activated egg. In this model, GNU localization would function to localize and activate PNG at a subset of cytoplasmic RNPs where the most imperative targets are present. Although our data are more consistent with the 'beacon' model, we cannot presently rule out the 'sequestration' model. Experiments using FRAP to measure how strongly GNU is bound to RNP granules could be helpful, as sequestration would require tight retention of GNU in the granules.

A final implication of this work is the possible roles of GNU in addition to the activation of PNG. Whether GNU regulation of PNG follows the 'sequestration' or 'beacon' models, the presence of GNU in oocytes suggests a potential PNG-independent role in regulating maternal mRNA translation. This idea is further supported by the effect of *gnu* on BIC-C RNP granules. A detailed analysis of mRNA translation in *gnu* null mutant oocytes, as well as in *gnu*[9A] and *gnu*[ΔSAM] mutant mature oocytes, is needed to assess the independent roles of GNU in regulating RNP granules and mRNA translation in mature oocytes.

Understanding the regulation of the PNG kinase complex, a major regulator of mRNA translation, and the role of its subunits is crucial for understanding the transition from oocyte to embryo. Moreover, in addition to implications in fertility, understanding the relationship between the PNG complex, RNP granules, and mRNA translation will yield insights into conserved regulation underlying developmental transitions that require rapid control of translation. Indeed, parallels between the roles of GNU and the PNG complex can be found in other organisms and in other developmental contexts. During left-right patterning in frog and fish, BICC1, the BIC-C homolog, can bind to the ankyrin-repeat proteins ANKS3 and ANKS6 (*Rothé et al., 2018*; *Rothé et al., 2020*). The interaction with ANKS3 can recruit a Ser/Thr kinase, NEK7, to regulate left-right patterning by restricting its activity to the right side of the animal (*Ramachandran et al., 2015*; *Rothé et al., 2020*). This model of regulation of NEK7 by ANKS3 and BICC1 is reminiscent of our model of regulation of PNG activity by GNU and BIC-C. The interaction with ANKS3 and ANKS6 also plays a role in regulating BICC1 polymerization and stabilization (*Rothé et al., 2020*). Interestingly, in addition to the ankyrin repeats, ANKS6 and ANKS3 also contain a SAM domain, and they can interact with BICC1 via SAM:SAM domain interactions (*Bakey et al., 2015*; *Rothé et al., 2018*). In the case of the PNG complex, a Ser/Thr kinase, these features are split between the two regulating subunits. Whereas GNU contains a SAM domain, in PLU the only discernable domains are ankyrin repeats (*Axton et al., 1994*). The similarities between the regulation of these two complexes suggest that similar control mechanisms are at play in different developmental contexts.

# Materials and methods

**Key resources table**

| Reagent type (species) or resource | Designation | Source or reference | Identifiers | Additional information |
|---|---|---|---|---|
| Gene (*Drosophila melanogaster*) | *gnu* | | FLYB: FBgn0001120 | |
| Gene (*D. melanogaster*) | *png* | | FLYB: FBgn0000826 | |
| Gene (*D. melanogaster*) | *plu* | | FLYB: FBgn0003114 | |
| Gene (*D. melanogaster*) | *Bic-C* | | FLYB: FBgn0000182 | |
| Gene (*D. melanogaster*) | *tral* | | FLYB: FBgn0041775 | |
| Gene (*D. melanogaster*) | *me31b* | | FLYB: FBgn0004419 | |
| Strain, strain background (*D. melanogaster*) | WT: *Oregon R* | N/A | | |
| Genetic reagent (*D. melanogaster*) | *gnu-wt-gfp[1-5]* | *Hara et al., 2017* | | Genotype: w;P{w[+mC], gnu-wt-gfp}; gnu[305]/ TM3, Sb |
| Genetic reagent (*D. melanogaster*) | *gnu-wt-gfp[1-4]* | *Hara et al., 2017* | | Genotype: w, P{w[+mC], gnu-wt-gfp};;gnu[305]/TM3, Sb |

*Continued on next page*

*Continued*

| Reagent type (species) or resource | Designation | Source or reference | Identifiers | Additional information |
|---|---|---|---|---|
| Genetic reagent (*D. melanogaster*) | gnu-wt-gfp[1-8] | *Hara et al., 2017* | | Genotype: w;P{w[+mC], gnu-wt-gfp}; gnu[305]/ TM3, Sb |
| Genetic reagent (*D. melanogaster*) | gnu-9A-gfp[2-7] | *Hara et al., 2017* | | Genotype: w;P{w[+mC], gnu-9A-gfp}; gnu[305]/ TM3, Sb |
| Genetic reagent (*D. melanogaster*) | gnu-9A-gfp[2-8] | *Hara et al., 2017* | | Genotype: w;P{w[+mC], gnu-9A-gfp}; gnu[305]/ TM3, Sb |
| Genetic reagent (*D. melanogaster*) | tral-gfp[89] | The Flytrap Project; *Morin et al., 2001*; PMID:11742088 | Flytrap:G00089; DGRC:110584; RRID:DGGR_110658 | Genotype: w[*];P{w[+mC] =PTT-un1} G00089 |
| Genetic reagent (*D. melanogaster*) | me31b-gfp | *Nakamura et al., 2001* | | Genotype: w, P{[W[+mC]], me31b-gfp} |
| Genetic reagent (*D. melanogaster*) | Bic-C-gfp[v318872] | *Sarov et al., 2016* | RRID:VDRC_v318872 | Transgene carries dsRed marker. Genotype: w;;PBac{fTRG01264.sf GFP-TVPTBF}VK00033 |
| Genetic reagent (*D. melanogaster*) | Bic-C-gfp[v318334] | *Sarov et al., 2016* | RRID:VDRC_v318334 | Transgene carries dsRed marker. Genotype: w;;PBac{fTRG01264.sf GFP-TVPTBF}VK00033 |
| Genetic reagent (*D. melanogaster*) | gnu-ΔSAM-gfp[3-2] | This paper | | Genotype: w;P{w[+mC], gnu-ΔSAM-gfp}; gnu[305]/ TM3, Sb |
| Genetic reagent (*D. melanogaster*) | gnu-ΔSAM-gfp[3-8] | This paper | | Genotype: w;P{w[+mC], gnu-ΔSAM-gfp}; gnu[305]/ TM3, Sb |
| Genetic reagent (*D. melanogaster*) | gnu-mKATE2 | This paper | | Genotype: w;P{w[+mC], gnu-mKATE2}; gnu[305]/ TM3, Sb |
| Genetic reagent (*D. melanogaster*) | BiC-C[4] | *Schüpbach and Wieschaus, 1991* | RRID:BDSC_3248 | Genotype: Bic-C[4], cn[1], exu[1], bw[1]/CyO |
| Genetic reagent (*D. melanogaster*) | Bic-C[PE37] | *Schüpbach and Wieschaus, 1991* | | Genotype: Bic-C[PE37], bw[1]/CyO |
| Genetic reagent (*D. melanogaster*) | gnu[305] | *Freeman and Glover, 1987* | FLYB: FBal0005121 | Genotype: ru, gnu[305], th, st, roe, p(p), e(s), ca/TM3, Sb |
| Genetic reagent (*D. melanogaster*) | png[1058] | *Shamanski and Orr-Weaver, 1991*; PMID:1913810 | RRID:BDSC_38437 | Genotype: y[1], png[1058], w[*]/FM6 |
| Genetic reagent (*D. melanogaster*) | h2av-gfp | | RRID:BDSC_24163 | Genotype: w[*]; P{w[+mC] =His2Av-EGFP.C}2/ SM6a |
| Genetic reagent (*D. melanogaster*) | pdi-gfp | | RRID:BDSC_6839 | Genotype: w[1118]; P{w[+mC]=PTT-GA}Pdi[G00198]/TM3, Sb[1] Ser[1] |
| Genetic reagent (*D. melanogaster*) | me31b-gfp; Bic-C[305]/Xa | This paper | | Generated from cross between me31b-gfp females and *Bic-C[4]* males. Genotype: w, P{[W[+mC]], me31b-gfp}; Bic-C[4]/Xa |

*Continued on next page*

*Continued*

| Reagent type (species) or resource | Designation | Source or reference | Identifiers | Additional information |
|---|---|---|---|---|
| Genetic reagent (*D. melanogaster*) | *gnu-gfp/+; Bic-C[305]/Sco* | This paper | | Generated from cross between gnu-wt-gfp[1-4] females and Bic-C[4] males. Genotype: *w;P{w[+mC], gnu-wt-gfp}/+; Bic-C[4]/Sco* |
| Genetic reagent (*D. melanogaster*) | *me31b-gfp;; gnu[305]/TM3* | This paper | | Genotype: *w, P{[W[+mC]], me31b-gfp};; gnu[305]/TM3* |
| Genetic reagent (*D. melanogaster*) | *png[1058]/ FM0; gnu-wt-gfp; gnu[305]/TM3* | This paper | | Genotype: *png[1058]/FM0; P{w[+mC], gnu-wt-gfp}/+; gnu[305]/TM3* |
| Genetic reagent (*D. melanogaster*) | *gnu-gfp/+; BicC[4]/Bic-C[PE37]* | This paper | | Generated from genetic cross between *gnu-gfp/+; BicC[4]/Sco* males and *Bic-C[PE37]* females. |
| Genetic reagent (*D. melanogaster*) | *plu-gfp[5-04]* | This paper | | PLU-GFP transgene; good expression; rescues homozygous *plu*. Genotype: *y,w; P{plu C/ GFP E.1}* |
| Genetic reagent (*D. melanogaster*) | *png-gfp[F-041]* | This paper | | PNG-GFP transgene; good expression; rescues homozygous *png[1058]*. Genotype: *y,w; P{png-gfp}* |
| Genetic reagent (*D. melanogaster*) | *twine/CyO* | Other | RRID:BDSC_4274; RRID:Kyoto_107663 | Sterile males due to spermlessness. Genotype: *twe[1] cn[1] bw[1]/CyO* |
| Antibody | Guinea pig polyclonal anti-GNU | *Lee et al., 2003* | PMID:14665672 | (1/5000) in TBS-T |
| Antibody | Rabbit polyclonal anti-BIC-C | P. Lasko (McGill University) | | (1/2000) in TBS-T |
| Antibody | Guinea pig polyclonal anti-GFP | M. Pardue (MIT) | | (1/5000) in TBS-T |
| Antibody | Rat monoclonal anti-αTUB YOL1/34 | AbD Serotec (Bio-Rad) | | (1/1000) in TBS-T |
| Antibody | Mouse monoclonal anti-CycB | Developmental Studies Hybridoma Bank | DSHB Cat#F2F4 RRID:AB_528189 | (1/100) in Hikari Solution B |
| Antibody | Rat monoclonal anti-MBP | Sigma-Aldrich | Sigma-Aldrich: SAB4200082 | (1/2000) in Hikari Solution |
| Antibody | Rabbit polyclonal anti-GST labeled with HRP | MBL | MBL:PM013-7; RRID:AB_10598029 | (1/5000) in Hikari solution |
| Antibody | Rabbit polyclonal anti-ME31B | *Nakamura et al., 2001* | PMID:27791980; RRID:AB_2568986 | (1/10,000) in TBS-T |
| Antibody | Donkey polyclonal HRP-conjugated anti-guinea pig IgG | Jackson Immuno Research | Jackson Immuno Research: 706-035-148: RRID:AB_2340447 | |
| Antibody | Goat polyclonal HRP-conjugated anti-mouse IgG | Jackson Immuno Research | Jackson Immuno Research: 115-035-164: RRID:AB_2338510 | |
| Antibody | Goat polyclonal HRP-conjugated anti-rat IgG | Jackson Immuno Research | Jackson Immuno Research:112-035-062: RRID:AB_2338133 | |

*Continued on next page*

*Continued*

| Reagent type (species) or resource | Designation | Source or reference | Identifiers | Additional information |
|---|---|---|---|---|
| Antibody | Donkey polyclonal HRP-conjugated anti-rabbit IgG | Jackson Immuno Research | Jackson Immuno Research: 711-035-152; RRID:AB_10015282 | |
| Antibody | Recombinant Nanobody gba488-100 (GFP-Booster Atto488) | Chromotek | | Booster for GFP fluorescence (1/400) in diluent buffer |
| Recombinant DNA reagent | pCasPeR4_gnu-wt-gfp | *Hara et al., 2017* | RRID:Addgene_113005 | |
| Recombinant DNA reagent | pCasPeR4_gnu-ΔSAM-gfp | This paper | | Generated from pCas PeR4_gnu-wt-gfp |
| Recombinant DNA reagent | pCasPeR4_gnu-mkate2 | This paper | | Generated from pCas PeR4_gnu-wt-gfp |
| Commercial assay or kit | gtm_20 anti-GFP magnetic beads | Chromotek | | GFP immunoprecipitation |
| Chemical compound, drug | HIKARI signal enhancer | Nacalai | Nacalai: 02270–81 | Signal Enhancer HIKARI for Western Blotting and ELISA |
| Software, algorithm | Imaris (Bitplane) | Bitplane | | Imaging data visualization and analysis |
| Software, algorithm | FIJI (ImageJ) | ImageJ | | Imaging data visualization and analysis |
| Software, algorithm | Scaffold (version 4) | Proteome Software | | Proteomic data visualization and analysis |

## Fly stocks and transgenic lines

All flies were fed a cornmeal and molasses diet and kept at 18, 22, or 25°C. *png*[1058], *gnu*[305], *BiC-C*[4], and *Bic-C*[PE37] have been described (*Shamanski and Orr-Weaver, 1991*; *Fenger et al., 2000*; *Freeman and Glover, 1987*; *Schüpbach and Wieschaus, 1991*). Transgenic *gnu*[wt]*-gfp*, *gnu*[9A]*-gfp*, *tral-gfp*[89], and *me31b-gfp* have also been described (*Hara et al., 2017*; *Morin et al., 2001*; *Nakamura et al., 2001*). The *Bic-C-gfp*[v318872] stock (*Sarov et al., 2016*) was obtained from the Vienna *Drosophila* Resource Center (Vienna, Austria). *Oregon R* was used as a no transgene control.

For the SAM domain deletion mutant, the *gnu*[ΔSAM] sequence was swapped for *gnu*[wt] in a pGEM T-easy plasmid containing the endogenous *gnu* sequence followed by a *gfp* sequence and a linker as described in *Hara et al., 2017*. The resulting *gnu* region was cloned into pCasPeR4 that had been digested with *Bam*HI and *Eco*RI, using Gibson Assembly Master Mix (NEB, Ipswich, MA). For the *gnu-mKATE2* transgenic line, the *gfp* sequence was swapped for an *mKATE2* sequence prior to cloning into the pCasPeR4 plasmid. pCasPeR4 *gnu*[ΔSAM]*-gfp* and pCasPeR4 *gnu-mkate2* were injected into *w*[1118] embryos and transgenics were recovered by P-element transposition (BestGene, Inc, Chino Hills, CA).

## Immunoprecipitation of GNU from mature oocytes and mass spec analysis

Mature stage 14 oocytes were isolated in Grace's Unsupplemented Insect Medium (Life Technologies, Carlsbad, CA) from female flies expressing a *gnu*[wt]*-gfp*, *gnu*[9A]*-gfp*, or *gnu*[ΔSAM]*-gfp* transgene, washed with embryo wash buffer (1× phosphate-buffered saline [PBS], 0.2% BSA, and 0.1% Triton X-100) and frozen in liquid nitrogen. The oocytes were homogenized in NP-40 lysis buffer (50 mM Tris-HCl pH 8.0, 150 mM NaCl, 2.5 mM EGTA, 2.5 mM EDTA, 1% NP40, 1 mM DTT, and 1× complete EDTA-free protease inhibitor cocktail [Roche, Indianapolis, IN], 125 nM okadaic acid). After spinning at 10 krpm at 4°C for 10 min, supernatants were transferred to new tubes and the protein concentration adjusted to 1 mg/mL. GFP-tagged protein was immunoprecipitated with anti-GFP magnetic beads, GFP-Trap Magnetic Agarose (Chromotek, Planegg-Martinsried, Germany), for 1 hr

at 4°C. After incubation with beads, samples were washed three times with NP-40 lysis buffer with 300 mM NaCl. Pull-downs from no GFP and *h2av-gfp* transgenic oocytes were performed as controls.

For immunoprecipitation and immunoblots, extracts were prepared from 300 oocytes and immunoprecipitated with 5 µL of anti-GFP magnetic beads in a volume of 500 µL. Samples were resuspended in 40 µL 4× SDS buffer, boiled for 10 min, and separated by 10% SDS-PAGE.

For mass spectrometry, extracts were prepared from 1500 oocytes, and 40 µL of anti-GFP magnetic beads were used for immunoprecipitation in a total volume of 3 mL. Samples were eluted three times in 50 µL 0.2 M glycine and two times with 150 µL Elution buffer (50 mM Tris-HCl pH 8.0, 150 mM NaCl, 2.5 mM EGTA, 2.5 mM EDTA, 1 mM DTT, 1× complete EDTA-free protease inhibitor cocktail [Roche, Indianapolis, IN]). The eluate was then neutralized by the addition of 150 µL 2M Tris pH 8.5, flash frozen, and stored at –80°C.

For mass spec identification of proteins co-immunoprecipitated with GNU-GFP, protein digestion, chromatographic separation of peptides, mass spectrometry, and protein identification were done as described previously (*Hara et al., 2017*).

For identification of interactors with GNU, label-free quantitation and analysis were performed using the Scaffold software. Relative enrichment in GNU-GFP over a no GFP control was determined from the label-free quantitation values. Proteins with at least a sevenfold relative enrichment in GNU-GFP samples over the no GFP control were considered as positive hits. For comparison between GNU$^{WT}$-GFP and GNU$^{\Delta SAM}$-GFP, total spectrum counts for each protein were normalized to the total spectrum count for GNU in each sample. A multiple t-test analysis was then performed on the normalized rations to determine significance for each protein between the two samples. The comparison between GNU$^{WT}$-GFP and H2Av-GFP was performed similarly, except the total spectrum counts for each protein were normalized to the total spectrum count for GFP in each sample.

## Immunoblots

Oocyte isolation, protein extract preparation, SDS-PAGE, filter transfer, antibody binding, protein detection, and membrane re-probing were done as previously described (*Hara et al., 2017*). All immunoblot analyses were repeated in at least two biological replicates.

Primary antibodies used in this study were guinea pig anti-GNU (1/5000 in TBS-T) (*Lee et al., 2003*), rabbit anti-BIC-C (1/2000 in TBS-T; from Paul Lasko, McGill University), guinea pig anti-GFP (1/5000 in TBS-T; a gift from Mary-Lou Pardue, MIT), rabbit anti-ME31B (1/10,000 in TBS-T) (*Nakamura et al., 2001*), rat anti-αTUB YOL1/34 (1/1000 in TBS-T; AbD Serotec, Raleigh), mouse anti-CycB (1/100, TBS-T; Developmental Studies Hybridoma Bank, Iowa City, IA), rat anti-MBP (1/2000 in Hikari Solution; Sigma-Aldrich, St Louis, MO), and rabbit anti-GST labeled with HRP (1/5000 in Hikari solution; MBL, Woburn, MA). Secondary antibodies used were HRP-conjugated anti-rabbit IgG, anti-guinea pig IgG, anti-mouse IgG, and anti-rat IgG (Jackson ImmunoResearch, West Grove, PA).

## Localization experiments and imaging

For co-localization experiments, *gnu-mkate2* females were crossed to males carrying the desired *gfp* transgene, and F1 female progeny were used for the imaging experiment.

To examine GNU-GFP localization in *Bic-C* mutants, *gnu$^{wt}$-gfp* females were crossed to a *w; Sco/CyO* balancer stock, and *gnu$^{wt}$-gfp/+; Sco/+* females were recovered. These females were then crossed to either *Bic-C$^4$/CyO* or *Bic-C$^{PE37}$/CyO* males, and the resulting *gnu$^{wt}$-gfp/+; Bic-C$^4$/CyO* or *gnu$^{wt}$-gfp/+; Bic-C$^{PE37}$/CyO* siblings crossed to create stable lines. For recovery of *gnu$^{wt}$-gfp/+; Bic-C$^{PE37}$/ Bic-C$^4$*, crosses were performed between *gnu$^{wt}$-gfp/+; Bic-C$^4$/CyO* and *gnu$^{wt}$-gfp/+; Bic-C$^{PE37}$/CyO* and females of the desired genotype collected.

To analyze BIC-C-GFP localization in *gnu* mutants, recombinant *Bic-C-gfp* and *gnu$^{305}$/TM6* stocks were generated.

Isolated mature stage 14 oocytes were hand dissected from fattened females of each desired genotype in Grace's Insect Medium, unsupplemented (Life Technologies, Carlsbad, CA). For imaging of oocytes, isolated mature oocytes were fixed in 4% formaldehyde as described in *Page and Orr-Weaver, 1997*. The vitelline membrane was removed manually by rolling between two glass slides. Oocytes were then incubated with a 1:400 dilution of GFP-booster Atto488 (Chromotek, Inc,

Islandia, NY) and stained with Hoechst 33342 (Thermo Fisher Scientific, Inc, Waltham, MA). Immuno-fluorescence samples were scored on a Zeiss LSM 710 Laser Scanning Confocal with Plan Apochromat 63× objective. Images were analyzed with ImageJ software.

Embryos were collected for 1 or 2 hr, dechorionated in 50% bleach, and washed with 1× PBS. For embryo imaging, embryos were fixed and stained with DAPI as described previously (*Shamanski and Orr-Weaver, 1991*). Embryos were imaged on a Nikon ECLIPSE Ti microscope with Plan Fluor 10× or Plan Apo 20× objectives. Images were analyzed with ImageJ software.

## In vitro egg activation

Stage 14 oocytes were activated as previously described (*Horner and Wolfner, 2008*; *Mahowald et al., 1983*; *Page and Orr-Weaver, 1997*). Stage 14 oocytes were isolated in isolation buffer from virgin females enriched for mature oocytes and activated in activation buffer. Activated oocytes were dechorionated and selected for successful activation by treatment with 50% bleach then washed with $H_2O$ and embryo wash buffer. For immunoblots, 8–30 activated eggs in a volume of 0.5 μL 1× PBS per egg in a 1.5 mL tube were frozen in liquid nitrogen and stored at –80°C.

## PNG kinase activation assay with GNU mutants

Recombinant PNG kinase complex was purified and the PNG kinase activity assay was performed as previously described (*Hara et al., 2017*) with modifications noted below. MBP-GNU was expressed and purified as described in *Hara et al., 2017*. The *gnu* ORF was cloned into pMAL-c2x (NEB, Ipswich, MA). The $gnu^{P17L}$ and $gnu^{\Delta SAM}$ mutant cDNAs were made using PCR and were cloned into pMAL-c2x (NEB, Ipswich, MA). The MBP-fused $GNU^{WT}$ and mutants were expressed in BL21 *Escherichia coli*, purified following manufacturer protocols (NEB, Ipswich, MA), and dialyzed in TBS with 0.05% NP-40 and 1 mM DTT.

Recombinant PNG kinase complex (PNG-FLAG, PLU-His, GNU) containing 6 ng PNG-FLAG was incubated with 20 pg of purified $MBP-GNU^{WT}$, $MBP-GNU^{\Delta SAM}$, or $MBP-GNU^{P17L}$ (a negative control) together with 6 μg of Myelin Basic Protein (a PNG kinase in vitro substrate) in 10 μL PNG reaction buffer (20 mM Tris-HCl pH 7.5, 3 mM $MnCl_2$, 10 mM $MgCl_2$, 80 mM disodium β-glycerophosphate, 100 mM ATP, 1 mM DTT, and 1× complete EDTA-free protease inhibitor cocktail [Roche, Indianapolis, IN]) with 7.4 MBq/mL [γ-$^{32}$P]ATP at 30°C for 15 min. About 5 μL 3× LSB with 25 mM EDTA was added. The samples were heated at 96°C for 5 min and separated on 15% SDS-PAGE. After CBB staining, phosphorylated Myelin Basic Protein was detected by autoradiography.

## GST-BIC-C pull-down

The *Bic-C* ORF was cloned into pGEX6P-1 (GE Healthcare, Waukesha, WI). GST-BIC-C protein was expressed in BL21 competent *E. coli* in 100 mL LB. Following PBS washes, the cells were resuspended in 2 mL PBS supplemented with 1× complete EDTA free protease inhibitor cocktail (Roche, Indianapolis, IN) and 1 mM DTT and lysed by sonication. After the addition of final 1% Triton X-100, the cell lysate was incubated on ice for 15 min and spun at 4°C for 15 min at 13 krpm to recover soluble proteins. The soluble proteins were added to 100 μL glutathione sepharose 4B (GE Healthcare, Waukesha, WI) at room temperature for 30 min. The GST-BIC-C bound beads were washed with 1 mL PBS with 1× Complete EDTA free protease inhibitor cocktail (Roche, Indianapolis, IN) and 1% Triton X-100 three times, followed by three washes with 1 mL pull-down buffer (150 mM NaCl, 50 mM Tris-HCl pH 8.0, 2.5 mM EGTA, 1% NP-40, 80 mM disodium β-glycerophosphate, 25 mM NaF, 1× complete EDTA free protease inhibitor cocktail [Roche, Indianapolis, IN], 1 mM DTT, 100 mM Sucrose, and 100 μg/mL BSA). About 10 μL of GST-BIC-C beads were incubated with MBP, $MBP-GNU^{WT}$, or $MBP-GNU^{\Delta SAM}$ (0.2 μg) in 20 μL pull-down buffer on ice for 1 hr with gentle mixing. The beads were washed with 1 mL pull-down buffer three times, added 10 μL 2× LSB, and heated at 96°C for 5 min. The samples were separated on 7.5% SDS-PAGE and proteins were detected by immunoblot using anti-MBP or anti-GST antibody.

## Acknowledgements

David Bartel generously hosted EA-P in his laboratory during part of this research. The authors thank Eric Spooner for mass spec analyses and Sebastian Lourido for statistical analysis of the results. The authors are grateful to Paul Lasko and Mary-Lou Pardue for antibodies, and to Adam Martin and

Trudi Schupbach for *Drosophila* stocks. Mariana Wolfner, David Bartel, Peter Reddien, and Adam Martin provided helpful comments on the manuscript. The authors thank Boryana Petrova, Jarrett Smith, and other members of the Bartel and Orr-Weaver labs for many helpful discussions. This work was supported by NIH grant GM118090 to TO-W and by a JSPS Postdoctoral Fellowship for Research Abroad, an Uehara Memorial Foundation Research fellowship, and JSPS KAKENHI Grant Number JP20H05367 to MH.

## Additional information

### Funding

| Funder | Grant reference number | Author |
|---|---|---|
| National Institutes of Health | GM118090 | Terry L Orr-Weaver |
| JSPS | | Masatoshi Hara |
| Uehara Memorial Foundation | | Masatoshi Hara |
| JSPS | JP20H05367 | Masatoshi Hara |

The funders had no role in study design, data collection and interpretation, or the decision to submit the work for publication.

### Author contributions

Emir E Avilés-Pagán, Conceptualization, Data curation, Formal analysis, Validation, Investigation, Visualization, Methodology, Writing - original draft, Writing - review and editing; Masatoshi Hara, Funding acquisition, Validation, Investigation, Methodology, Writing - review and editing; Terry L Orr-Weaver, Conceptualization, Supervision, Funding acquisition, Investigation, Methodology, Project administration, Writing - review and editing

### Author ORCIDs

Emir E Avilés-Pagán (iD) http://orcid.org/0000-0003-1245-0941
Masatoshi Hara (iD) https://orcid.org/0000-0001-8433-1111
Terry L Orr-Weaver (iD) https://orcid.org/0000-0002-7934-111X

### Decision letter and Author response

Decision letter https://doi.org/10.7554/eLife.67294.sa1
Author response https://doi.org/10.7554/eLife.67294.sa2

## Additional files

### Supplementary files

• Supplementary file 1. Supplementary table showing interactors with GNU in mature oocytes identified by IP-MS in ranked order. The fold enrichment in GNU-GFP over a no GFP control is shown. The source data for this table are in *Figure 1—figure supplement 1—source data 1*.

• Transparent reporting form

### Data availability

All data generated or analysed during this study are included in the manuscript and supporting files. Source data files have been provided for Figure 1, Figure 1-Supplement 1, Figure 1-Supplement 2, Figure 1-Supplement 3, Figure 2, Figure 3-Supplement 1, Figure 4, Figure 4-Supplement 1, Figure 5, Figure 5-Supplement 1, Figure 6, Table 1, and Supplementary file 1.

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
