## [Decision Letter]

**Acceptance summary:**

This interesting study addresses translational control of cell cycle regulators in the *Drosophila* oocyte. Using *Drosophila* genetics, a variety of biochemical tests, and microscopy, the authors show that GNU, an activating subunit of the kinase that regulates maternal mRNA translation at the oocyte-to-embryo transition, associates with ribonucleoprotein particles. The authors provide insights into the significance of this association for regulating the activity of the kinase.

**Decision letter after peer review:**

Thank you for submitting your article "The GNU subunit of PNG kinase, the developmental regulator of mRNA translation, an RNP component in *Drosophila* oocytes" for consideration by *eLife*. Your article has been reviewed by 3 peer reviewers, including Jon Pines as the Reviewing Editor and Reviewer #1, and the evaluation has been overseen by Kevin Struhl as the Senior Editor. The following individual involved in review of your submission has agreed to reveal their identity: Ilan Davis (Reviewer #3).

Essential revisions:

All the referees have agreed that only textual revisions are required for publication as follows:

The title of the manuscript would be improved if it were a statement of the key theme or finding.

Figure 1B: Although it looks as if less BIC-C is brought down with the GNU 9A mutant, it also looks as if less GNU is present in the IPs. These data should be quantified and normalised to the amount of GNU protein.

Figure 2B. Would benefit from a high magnification inset of the giant nuclei. What are they? Multiple aggregated nuclei?

Figure 4: It would be helpful to have high magnification insets for the merged images. The colour is also difficult to see in the merged images.

Figure 5: It would be helpful to have high magnification insets for each panel to illustrate the pattern of co-localisation. The colour is also difficult to see in the merged images.

Figure 6: The data do argue against a sequestration model for the regulation of PNG activity. One missing piece of evidence that could be helpful is how strongly GNU is bound to RNP granules, which could be estimated by FRAP. The authors could discuss this possibility.

In Figure 6A, lane 5, why are there two bands for GNU-GFP? Also, if not noted in the text it may be worth noting that band size differences for both GNU and BIC-C in lane 6 vs. 5, and BIC-C in 3 vs. 2, are due to changes in phosphorylation states, if this is the case.

The Discussion could be shortened, and a few other places (e.g. line 324 vs. 329) tightened, by removing wording repetitions.

In line 615-616 it may be worth citing Niepielko et al. Curr. Biol. 2018, which reports heterogeneity in RNP granules, though admittedly at the RNA rather than protein level.

Typos: BIC-C is missing its hyphen in panel 2A, and I think the crossed-out PLU should be removed from line 805.

*Reviewer #1:*

In this study the authors set out to determine how the GNU-PNG complex that regulates the translation of key cell cycle components at fertilisation is itself controlled. The study is careful, the data are clear and convincing, and the authors support their initial biochemistry with genetic and cell biological experiments. In my opinion this is a strong study with no major weaknesses in the data presented.

The conclusion that GNU is bound to BIC-C in RNP particles in the oocyte, and that its dispersal at fertilisation is relevant to the control of the translation of Cyclin B will be of interest because it reveals a feedback between RNP granules and the regulator of a kinase that controls mRNA unmasking and translation. The data also reveal that the make-up of RNA granules is heterogeneous. Lastly, the relationship revealed here could also have wider implications for kinase regulation in other developmental contexts.

Figure 1B: Although it looks as if less BIC-C is brought down with the GNU 9A mutant, it also looks as if less GNU is present in the IPs. These data should be quantified and normalised to the amount of GNU protein.

Figure 4: It would be helpful to have high magnification insets for the merged images.

Figure 5: It would be helpful to have high magnification insets for each panel to illustrate the patter on localisation.

Figure 6: The data do argue against a sequestration model for the regulation of PNG activity. One missing piece of evidence that could be helpful is how strongly GNU is bound to RNP granules, which could be estimated by FRAP.

*Reviewer #2:*

Avilés-Pagán et al. report that the *Drosophila* GNU protein, a subunit of the PNG kinase that regulates mRNA translation at the egg-to-embryo transition, associates with RNP particles in the mature oocyte. Upon egg activation, GNU disperses into the cytoplasm, concomitant with dispersal of the RNP particles. The authors' immunoprecipitation mass spectrometry, verified by immunoprecipitation and consistent with yeast two-hybrid results where available, show that GNU associates with several, but not all, proteins found in RNPs. They then determine many specifics of this association. GNU is shown to associate with BIC-C, and the RNP particles that contain GNU also contain BIC-C, defining a subpopulation of RNP particles. The authors show that GNU's SAM domain is needed to interact with BIC-C, and that mutations that disable GNU's CDK1 target sites diminish its interactions with RNP proteins. They propose two models for the implications of GNU's association with RNP particles. In one, GNU is sequestered from PNG by being stored in these particles. Its release upon egg activation frees GNU to complex with and activate PNG. Some of the data presented are consistent with this model, but others are decidedly not. This led the authors to propose and favor a new, creative, "beacon" model in which PNG is attracted to the GNU-containing subset of RNP particles, allowing the kinase to be activated there and to phosphorylate its targets there. The authors' data are consistent with this model.

The study is at the intersection between, and contributes to, several timely and important questions: the roles of RNP particles, IDP domain proteins, and mechanisms for transitioning to totipotency. The methods are state of the art. The experiments include all appropriate controls, appropriate sample sizes and biological repeats. The data are high-quality and clear, and interpreted carefully and rigorously. Conclusions are also verified by independent alternative approaches. As typical for this lab's papers, every time I thought of an alternative explanation for a result, the next sentence of the paper and next panel of the figure addressed my issue and laid it to rest. The interesting models are important for the research community to think about. The writing and logic are very clear. The only small concern, which does not diminish my enthusiasm for this work, is that some of the largest interpretations are speculative, including that the beacon model is not yet proven. However, this is already a very large paper, and there is not room for all possible tests. And the important question, rigorous and interesting results, and novel conclusions will be of great interest to the field.

Suggested modifications.

1. I think the Discussion could be shortened, and a few other places (e.g. line 324 vs. 329) tightened, by removing wording repetitions.

2. In Figure 6A, lane 5, why are there two bands for GNU-GFP? Also, if not noted in the text (I may have missed it) it may be worth noting that band size differences for both GNU and BIC-C in lane 6 vs. 5, and BIC-C in 3 vs. 2, are due to changes in phosphorylation states, if this is the case.

3. In line 615-616 it may be worth citing Niepielko et al. Curr. Biol. 2018, which reports heterogeneity in RNP granules, though admittedly at the RNA rather than protein level.

4. Typos: BIC-C is missing its hyphen in panel 2A, and I think the crossed-out PLU should be removed from line 805.

*Reviewer #3:*

The manuscript by Avilés-Pagán et al. follows on from their previous work (Hara et al. *eLife* 2018) in which they identified a relationship between Tral and GNU in the oocyte-embryo transition. This is an important time in development, at which the pre-pattern in the oocyte transitions to the beginning of bringing that pre-pattern to life using a combination of pre-localised determinants and new transcription in the early embryo. The current manuscript uses biochemical methods combined with microscopy in wild type and mutants to further characterise the interactions and relationships of the factors required for this transition. They examined the interactions of PGN and GNU (and requirement of its SAM domain for the interactions) as well as the co-localisation at granules. They found interesting interactions with GNU with several granule factors of major known developmental significant, TRAL, ME31B, and BIC-C in mature oocytes. The authors propose different classes of granules with distinct compositions and present two interesting alternative models for the regulation of PGN by GNU during the three critical developmental stages: mature oocyte, during activation and activated embryo.

There are some specific issues that I would recommend resolving, which in my opinion will greatly improve the manuscript and the clarity of data presentation.

1) I think the title of the manuscript would be improved if it were a statement of the key THM or finding. "The GNU subunit of PNG kinase, the developmental regulator of mRNA translation, an RNP component in *Drosophila* oocytes". What about GNU?

2) Figure 2B. Would benefit from a "blow up" inset of the giant nuclei. What are they? Multiple aggregated nuclei?

3) The merged (multi-channel in colour) images in Figures 4 and 5, are really unclear at least in the online reproduction. There is nearly any colour content in them (they are mostly black and white with a tinny hint of colour). This may just be a simple reproduction issue. It would also be good to present a bit more analysis or "blow up" of the granules to describe/quantitate the co-localisations presented.

---

## [Author Response]

Essential revisions:All the referees have agreed that only textual revisions are required for publication as follows:The title of the manuscript would be improved if it were a statement of the key theme or finding.

The title has been changed as suggested.

Figure 1B: Although it looks as if less BIC-C is brought down with the GNU 9A mutant, it also looks as if less GNU is present in the IPs. These data should be quantified and normalised to the amount of GNU protein.

We previously had included the quantitation of these data in the Source Data. Now in addition we mention it in the Results section and have included a supplemental figure with a graph showing the quantitation normalized to the amount of GNU protein (Figure 1 supplement 3).

Figure 2B. Would benefit from a high magnification inset of the giant nuclei. What are they? Multiple aggregated nuclei?

The gnu giant nuclei phenotype has been published many times over the past 30 years. As noted in the Results, the phenotype is the consequence of DNA replication in the absence of nuclear division, and as ploidy of the polar body and pronuclei increases they tend to fuse into a single, multi-lobed nucleus. We have added sentences to the figure legend reviewing these published data and cited in the legend two papers that extensively present and discuss the gnu mutant phenotype.

Figure 4: It would be helpful to have high magnification insets for the merged images. The colour is also difficult to see in the merged images.

We have included the suggested high magnification insets for the images in Figure 4 and Figure 4 Supplement 1. The green and magenta are more visible at the higher magnification.

Figure 5: It would be helpful to have high magnification insets for each panel to illustrate the pattern of co-localisation. The colour is also difficult to see in the merged images.

We have included the suggested high magnification insets for the images for Figure 5A. The green and magenta are more visible at the higher magnification.

Figure 6: The data do argue against a sequestration model for the regulation of PNG activity. One missing piece of evidence that could be helpful is how strongly GNU is bound to RNP granules, which could be estimated by FRAP. The authors could discuss this possibility.

This is now mentioned in the Discussion.

In Figure 6A, lane 5, why are there two bands for GNU-GFP? Also, if not noted in the text it may be worth noting that band size differences for both GNU and BIC-C in lane 6 vs. 5, and BIC-C in 3 vs. 2, are due to changes in phosphorylation states, if this is the case.

In immunoprecipitations of GNU-GFP from mature oocytes we often see a degradation product detectable on immunoblots with the anti-GFP antibody. We have marked this band with an asterisk in the figure and explain this in the figure legend.

Although it is possible that BIC-C phosphorylation states are changed during egg activation, we have not done experiments treating extracts with phosphatases that would permit us to test this. In addition, the extracts in the experiment in Figure 6A were not prepared with the amount of phosphatase inhibitors used in Hara et al., 2017 that would be necessary for us to make a conclusion about GNU phosphorylation.

The Discussion could be shortened, and a few other places (e.g. line 324 vs. 329) tightened, by removing wording repetitions.

We have condensed the Discussion, both removing word repetitions but also eliminating sentences that were repetitive in different paragraphs or too speculative.

In line 615-616 it may be worth citing Niepielko et al. Curr. Biol. 2018, which reports heterogeneity in RNP granules, though admittedly at the RNA rather than protein level.

This paper has been cited as suggested.

Typos: BIC-C is missing its hyphen in panel 2A, and I think the crossed-out PLU should be removed from line 805.

These typos have been corrected.